**Pollen-based quantitative land-cover reconstruction for northern Asia covering**
**the last 40 ka**
Xianyong Cao[a*], Fang Tian[a], Furong Li[b], Marie-José Gaillard[b], Natalia Rudaya[a,c,d],
Qinghai Xu[e], Ulrike Herzschuh[a,d,f*]
*[a] Alfred Wegener Institute for Polar and Marine Research, Research Unit Potsdam, Telegrafenberg A43, 14473*
*Potsdam, Germany*
*[b] Department of Biology and Environmental Science, Linnaeus University, Kalmar SE-39182, Sweden*
*[c] Institute of Archaeology and Ethnography, Siberian Branch, Russian Academy of Sciences, pr. Akad. Lavrentieva*
*17, Novosibirsk 630090, Russia*
*[d] Institute of Environmental Science and Geography, University of Potsdam, Karl-Liebknecht-Str. 24, 14476*
*Potsdam, Germany*
*[e] College of Resources and Environment Science, Hebei Normal University, Shijiazhuang 050024, China*
*[f] Institute of Biochemistry and Biology, University of Potsdam, Karl-Liebknecht-Str. 24, Potsdam 14476, Germany*
**ABSTRACT**
We collected the available relative pollen productivity estimates (PPEs) for 27 major
pollen taxa from Eurasia and applied them to estimate plant abundances during the
last 40 cal. ka BP (calibrated thousand years before present) using pollen counts from
203 fossil pollen records in northern Asia (north of 40°N). These pollen records were
organised into 42 site-groups, and regional mean plant abundances calculated using
the REVEALS (Regional Estimates of Vegetation Abundance from Large Sites)

∗ Corresponding authors. Alfred Wegener Institute Helmholtz Centre for Polar and Marine Research, Research Unit Potsdam, Telegrafenberg A43, Potsdam 14473, Germany. X. Cao (xcao@itpcas.ac.cn); U. Herzschuh (Ulrike.Herzschuh@awi.de). Present address for Xianyong Cao: Key Laboratory of Alpine Ecology, CAS Center for Excellence in Tibetan Plateau Earth Sciences, Institute of Tibetan Plateau Research, Chinese Academy of Sciences, Beijing 100101, China.

model. Time-series clustering, constrained hierarchical clustering, and detrended canonical correspondence analysis were performed to investigate the regional pattern, time, and strength of vegetation changes, respectively. Reconstructed regional plant-functional type (PFT) components for each site-group are generally consistent with modern vegetation, in that vegetation changes within the regions are characterized by minor changes in the abundance of PFTs rather than by increase in new PFTs, particularly during the Holocene. We argue that pollen-based REVEALS estimates of plant abundances should be a more reliable reflection of the vegetation as pollen may overestimate the turnover, particularly when a high pollen producer invades areas dominated by low pollen producers. Comparisons with vegetation-independent climate records show that climate change is the primary factor driving land-cover changes at broad spatial and temporal scales. Vegetation changes in certain regions or periods, however, could not be explained by direct climate change, for example inland Siberia, where a sharp increase in evergreen conifer tree abundance occurred at ca. 7–8 cal. ka BP despite an unchanging climate, potentially reflecting their response to complex climate–permafrost–fire–vegetation interactions and thus a possible long-term-scale lagged climate response.

*Keywords*: boreal forests, China, Holocene, late Quaternary, pollen productivity, quantitative reconstruction, Siberia, vegetation

**1 Introduction**

High northern latitudes such as northern Asia experience above-average temperature increases in times of past and recent global warming (Serreze et al., 2000; IPCC, 2007), known as polar amplification (Miller et al., 2010). Temperature rise is expected to promote vegetation change as the vegetation composition in these areas is assumed to be controlled mainly by temperature (Li J. et al., 2017; Tian et al., 2018). However, a more complex response can occur mainly because vegetation is not linearly related to temperature change (e.g. due to resilience, stable states or time-lagged responses; Soja et al., 2007; Herzschuh et al., 2016) and/or vegetation is only indirectly limited

by temperature while other temperature-related environmental drivers such as
permafrost conditions are more influential (Tchebakova et al., 2005).
Such complex relationships between temperature and vegetation may help explain
several contradictory findings of recent ecological change in northern Asia. For
example, simulations of vegetation change in response to a warmer and drier climate
indicate that steppe should expand in the present-day forest–steppe ecotone of
southern Siberia (Tchebakova et al., 2009) but, contrarily, pine forest has increased
during the past 74 years, probably because the warming temperature was mediated by
improved local moisture conditions (Shestakova et al., 2017). In another example,
evergreen conifers, which are assumed to be more susceptible to frost damage than
*Larix*, expanded their distribution by 10% during a period with cooler winters from
2001 to 2012, while the distribution of *Larix* forests decreased by 40% on the West
Siberian Plain as revealed by a remote sensing study (He et al., 2017). Additionally,
some field studies and dynamic vegetation models infer a rapid response of the
treeline to warming in northern Siberia (e.g., Moiseev, 2002; Soja et al., 2007;
Kirdyanov et al., 2012), but combined model- and field-based investigations of larch
stands in north-central Siberia reveal only a densification of tree-stands, not an areal
expansion (Kruse et al., 2016; Wieczorek et al., 2017).
These findings on recent vegetation dynamics that contradict a straightforward
vegetation-temperature relationship may be better understood in the context of
vegetation change over longer time-scales. Synthesizing multi-record pollen data is
the most suitable approach to investigate quantitatively the past vegetation change at
broad spatial and long temporal scales. Broad spatial scale pollen-based land-cover
reconstructions have been made for Europe (e.g. Mazier et al., 2012; Nielsen et al.,
2012; Trondman et al., 2015) and temperate China (Li, 2016) for the Holocene.
However, vegetation change studies in northern Asia are restricted to biome
reconstructions (Tarasov et al., 1998, 2000; Bigelow et al., 2003; Binney et al. 2017;
Tian et al., 2018), which do not reflect compositional change. Syntheses of pure
pollen percentage data are not appropriate due to differences in pollen productivity,
which may result in an overestimation of the strength of vegetation changes (Wang
and Herzschuh, 2011). This might be particularly severe when strong pollen producers
such as pine (Mazier et al., 2012) invade areas dominated by low pollen producers
such as larch (Niemeyer et al, 2015). Marquer et al. (2014, 2017) also demonstrated
the strength of pollen-based REVEALS estimates of plant abundance in studies of
Holocene vegetation change and plant diversity indices in Europe. Accordingly,
syntheses of quantitative plant cover derived from the application of PPEs to multiple
pollen records (Trondman et al., 2015; Li, 2016) should be a better way to investigate
Late Glacial and Holocene vegetation change in northern Asia.
In this study, we employ the taxonomically harmonized and temporally standardized
fossil pollen datasets available from eastern continental Asia (Cao et al., 2013, 2015)
and Siberia (Tian et al., 2018) covering the last 40 cal. ka BP (henceforth abbreviated
to ka). We compile all the available PPEs from Eurasia and use the mean estimate for
each taxon. Finally, we quantitatively reconstruct plant cover using the REVEALS
model (Sugita, 2007) for 27 major taxa at 18 key time slices. We reveal the nature,
strength, and timing of vegetation change in northern Asia and its regional
peculiarities, and discuss the driving factors of vegetation change.
**2 Data and methods**
*2.1. Fossil pollen data process*
The fossil pollen records were obtained from the extended version of the fossil pollen
dataset for eastern continental Asia containing 297 records (Cao et al., 2013, 2015)
and the fossil pollen dataset for Siberia with 171 records (Tian et al., 2017). For the
468 pollen records, pollen names were harmonized to genus level for arboreal taxa
while family level for herbaceous taxa, and age-depth models were re-established
using the Bayesian age-depth modelling (further details are described in Cao et al.,
2013). We selected 203 pollen records from lacustrine sediments (110 sites) and peat
(93 sites) north of 40°N, with chronologies based on ≥3 dates and <500 year/sample
temporal resolution generally, following previous studies (Mazier et al., 2012; Nielsen
et al., 2012; Fyfe et al., 2013; Trondman et al., 2015). Out of the 203 pollen records,
170 sites (83 from lakes, 87 from bogs) have original pollen counts, while in the other
33 sites only pollen percentages are available. Due to overall low site density, we
decided to include these data. The pollen counts were back calculated from
percentages using the terrestrial pollen sum indicated in the original publications.
Detailed information (including location, data quality, chronology reliability, and data
source) of the selected sites is presented in Appendices 1 and 2.

Table 1 Selected time windows.

| Time window (cal a BP) | Abbreviated name |
|---|---|
| -60~100 | 0 ka |
| 100~350 | 0.2 ka |
| 350~700 | 0.5 ka |
| 700~1200 | 1 ka |
| 1700~2200 | 2 ka |
| 2700~3200 | 3 ka |
| 3700~4200 | 4 ka |
| 4700~5200 | 5 ka |
| 5700~6200 | 6 ka |
| 6700~7200 | 7 ka |
| 7700~8200 | 8 ka |
| 8700~9200 | 9 ka |
| 9700~10200 | 10 ka |
| 10500~11500 | 11 ka |
| 11500~12500 | 12 ka |
| 13500~14500 | 14 ka |
| 19000~23000 | 21 ka |
| 23000~27000 | 25 ka |
| 36000~44000 | 40 ka |

We selected 18 key time slices for reconstruction (Table 1) to capture the general
temporal patterns of vegetation change during the last 40 ka, i.e. 40, 25, 21, 18, 14,
and 12 ka during the late Pleistocene and 1000-year resolution (i.e. 500-year time
windows around each millennium, e.g. 0.7–1.2 ka, 1.7–2.2 ka, etc.) during the
Holocene. For the 0 ka time slice, the ca. 150-year time window (<0.1 ka) was set to
represent the modern vegetation. Since few pollen records have available samples at
the 0 ka time slice, the 0.2 and 0.5 ka time slices covered a 250-year or 350-year time
window (0.1~0.35 ka and 0.35~0.7 ka, respectively) to represent the recent vegetation,

following the strategy and time windows implemented for Europe (Mazier et al., 2012; Trondman et al., 2015). For the last glacial period, even broader time windows were chosen to offset the sparsely available samples (Table 1). Pollen counts of all available samples within one time window were summed up to represent the total pollen count for each time slice. In this study, we selected 27 major pollen taxa (with available PPE values) that form dominant components in both modern vegetation communities and the fossil pollen spectra and reconstruct their abundances in the past vegetation (Table 2).

Table 2 Fall speed of pollen grains (FS) and mean relative pollen productivity estimate (PPE) with standard error (SE) for the 27 selected taxa. Plant-functional type (PFT) assignment is according to previous biome reconstructions (Tarasov et al., 1998, 2000; Bigelow et al., 2003; Ni et al., 2010).

| PFT | PFT description | pollen type | FS (m/s) | PPE (SE) |
|-----|-----------------|-------------|----------|----------|
| I | evergreen conifer tree | *Pinus* | 0.031 [1] | 9.629 (0.075) |
| I | evergreen conifer tree | *Picea* | 0.056 [1] | 2.546 (0.041) |
| I | evergreen conifer tree | *Abies* | 0.120 [1] | 6.875 (1.442) |
| II | deciduous conifer tree | *Larix* | 0.126 [1] | 3.642 (0.125) |
| III | boreal deciduous tree | *Betula*_tree *Betula*_undiff. | 0.024 [1] | 8.106 (0.125) |
| III | boreal deciduous tree | *Alnus*_tree *Alnus*_undiff. | 0.021 [1] | 9.856 (0.092) |
| III | boreal deciduous tree | *Corylus* | 0.025 [2] | 1.637 (0.065) |
| IV | temperature deciduous tree | *Quercus* | 0.035 [1] | 6.119 (0.050) |
| IV | temperature deciduous tree | *Fraxinus* | 0.022 [1] | 2.046 (0.105) |
| IV | temperature deciduous tree | *Juglans* | 0.037 [3] | 4.893 (0.221) |
| IV | temperature deciduous tree | *Carpinus* | 0.042 [1] | 5.908 (0.285) |
| IV | temperature deciduous tree | *Tilia* | 0.032 [2] | 1.055 (0.066) |
| IV | temperature deciduous tree | *Ulmus* | 0.032 [2] | 6.449 (0.684) |
| V | boreal shrub | *Betula*_shrub | 0.024 [1] | 1.600 (0.132) |
| V | boreal shrub | *Alnus*_shrub | 0.021 [1] | 6.420 (0.420) |
| V | boreal shrub | *Salix* | 0.034 [2] | 1.209 (0.039) |
| V | boreal shrub | Ericaceae | 0.034 [4] | 0.200 (0.029) |
| VI | arid-tolerant shrub and herb | *Ephedra* | 0.015 [8] | 0.960 (0.140) |
| VI | arid-tolerant shrub and herb | *Artemisia* | 0.014 [6] | 9.072 (0.176) |
| VI | arid-tolerant shrub and herb | Chenopodiaceae | 0.019 [6] | 5.440 (0.460) |
| VII | grassland and tundra forb | Poaceae | 0.035 [4] | 1.000 (0.000) |
| VII | grassland and tundra forb | Cyperaceae | 0.035 [5] | 0.757 (0.044) |

| VII | grassland and tundra forb | Asteraceae | 0.051 [7] | 0.465 (0.066) |
| VII | grassland and tundra forb | *Thalictrum* | 0.007 [8] | 3.855 (0.258) |
| VII | grassland and tundra forb | Ranunculaceae | 0.014 [9] | 2.900 (0.363) |
| VII | grassland and tundra forb | Caryophyllaceae | 0.028 [9] | 0.600 (0.050) |
| VII | grassland and tundra forb | Brassicaceae | 0.002 [3] | 4.185 (0.188) |

[1] Eisenhut (1961); [2] Gregory (1973); [3] Li et al. (2017); [4] Broström et al. (2004); [5] Sugita et al.
(1999); [6] Abraham and Kozáková (2012); [7] Broström et al. (2002); [8] Xu et al. (2014); [9] Bunting et
al. (2013).
*2.2 The REVEALS model setting*
The REVEALS model assumes the PPEs of pollen taxa are constant variables over the
target period, and requires parameter inputs including sediment basin radius (m), fall
speed of pollen grain (FS, m/s), and PPE with standard error (SE; Sugita, 2007). The
areas of the 110 lakes were obtained from descriptions in original publications and
validated by measurements on Google Earth. Their basin radii were back-calculated
from their areas assuming a circular shape. There are 83 large lakes (radius >390 m;
following Sugita, 2007) in our dataset with a fairly even distribution across the study
area (Fig. 1; Appendix 1), which helps ensure the reliability of the regional vegetation
estimations (Sugita, 2007; Mazier et al., 2012). Only 18 bogs have published
descriptions about their size and it is infeasible to measure them on Google Earth
because of unclear boundaries. A test-run showed that using different bog radii (i.e. 5
m, 10 m, 20 m, 50 m, 100 m, 200 m and 500 m) did not significantly affect the
REVEALS estimates (Appendix 3), hence a standard (moderate size) radius of 100 m
was set for all bogs.
We collected available PPEs for the 27 selected pollen taxa from 20 studies in Eurasia
(Appendix 4). We calculated the mean PPE from all available PPE values, but
excluded records with PPE ≤ SE (Mazier et al., 2012). We included these PPEs for
various species in the mean PPE calculation for their family or genus. For
simplification, we did not evaluate the values or select PPE values following
consistent criteria as was done in Europe (Mazier et al., 2012). Instead, we used the
original values from the studies included in Mazier et al. (2012) and added new PPE
values from Europe published since the synthesis of Mazier et al. (2012). SE of the
mean PPE was estimated using the delta method (Stuart and Ord, 1994). Fall speeds
for each of the 27 pollen taxa were retrieved from previous studies (Table 2).
The REVEALS model generally performs best with pollen records from large lakes,
although multiple pollen records from small lakes and bogs (at least two sites) can
also produce reliable results where large lakes are absent (Sugita, 2007; Trondman et
al., 2016). Here, due to the sparse distribution of available sites, we divided the 203
sites into 42 site-groups, based on criteria of geographic location, vegetation type
(vegetation zone map modified from Tseplyayev, 1961; Dulamsuren et al., 2005; Hou,
2001), climate (based on modern precipitation and temperature contours), and
permafrost (Brown, 1997) following the strategy of Li (2016), and the pollen data
within one site-group should be of similar components and temporal patterns. To
ensure the reliability of REVEALS estimates of plant cover, each group includes at
least one large lake or two small sites (small lakes or bogs; Fig. 1; Appendix 5).

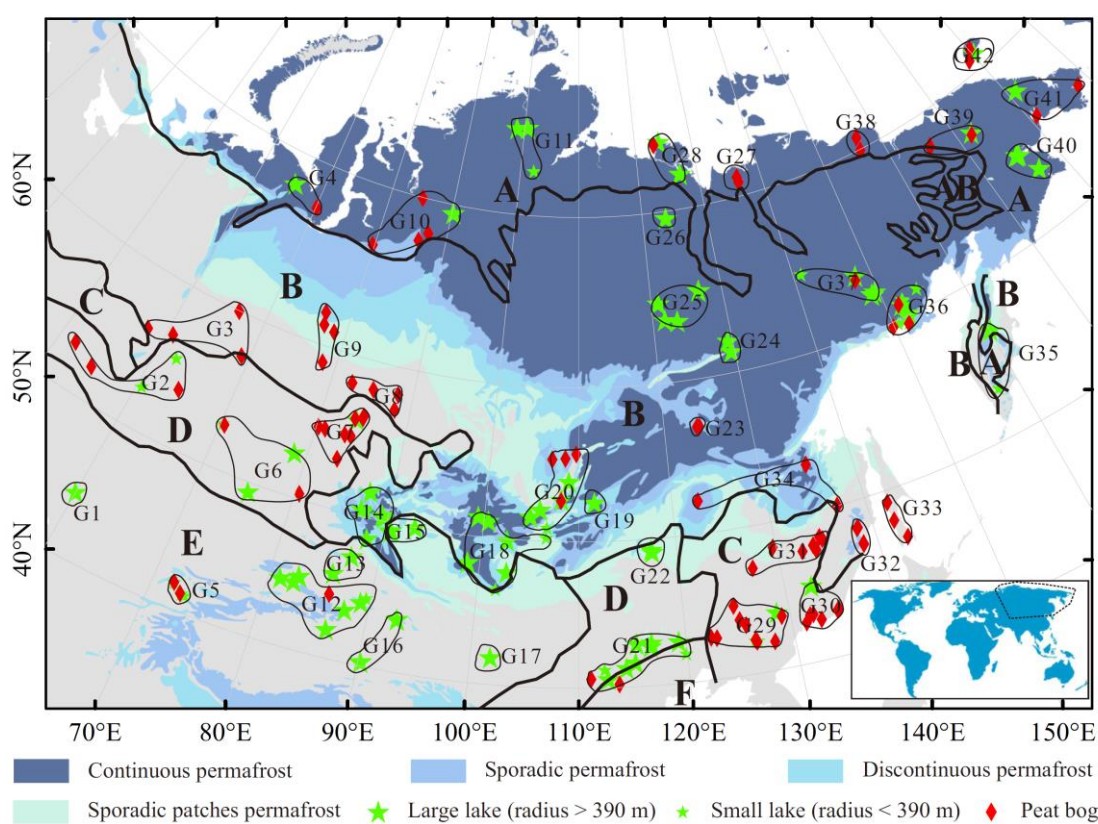

**Fig. 1.** Distribution of the 42 site-groups together with the modern vegetation zones and
permafrost extent in northern Asia. The vegetation-zone map modified from Tseplyayev (1961),
Dulamsuren et al. (2005), and Hou (2001) includes: A: tundra, B: taiga forest, C: temperate mixed
conifer-deciduous broadleaved forest, D: temperate steppe, E: semi-desert and desert; F:
warm-temperate deciduous forest.
The REVEALS model was run with a mean wind speed set to 3 m/s and neutral
atmospheric conditions following Trondman et al. (2015), and the maximum distance
of regional vegetation *Zmax* was set to 100 km. The lake and bog sites were
reconstructed using the models of pollen dispersal and deposition for lakes (Sugita,
1993) and bogs (Prentice, 1985), respectively in REVEALS version 5.0 (Sugita,
unpublished). The mean estimate of plant abundances from lakes and bogs was
calculated for each of the 42 site-groups, which includes both sediment types (using
the computer program bog.lake.data.fusion, Sugita, unpublished). Finally, the 27 taxa
were assigned to seven plant functional types (PFT; Table 1) following the PFT
definitions for China and Siberia (Tarasov et al. 1998, 2000; Bigelow et al. 2003; Ni
et al., 2010; Tian et al., 2017), with the restriction that each pollen taxon is attributed
to only one PFT according to the strategy of Li (2016) (Table 2).
*2.2 Numerical analyses of reconstruction*
The abundance variations of the seven PFTs during the Holocene (time slices between
12 and 1 ka) from 36 site-groups were used in a clustering analysis. Six site-groups
had to be excluded from the analysis due to poor coverage of time slices (G1, G5,
G17, G19, G27, G42). For site-groups with <3 missing time slices during the
Holocene (G3, G16, G26, G32, G33, G35, G38, G39, G41), linear interpolation was
employed to estimate the PFT abundances for the missing time slices. Time-series
clustering for the three-way dataset was performed to generate a distance matrix
among the site-groups using the *tsclust* function in the *dtwclust* package
(Sarda-Espinosa, 2018) in R 3.4.1 (R Core Team, 2017). The distance matrix was
employed in hierarchical clustering (using the *hclust* function in R) to cluster the
site-groups. Constrained hierarchical clustering (using *chclust* function in *rioja*
package version 0.9-15.1; Juggins, 2018) was used to determine the timing of primary
vegetation changes (i.e. the first split) in each site-group. A change was considered to
be significant when the split passed the broken-stick test. The amount of PFT
compositional change (turnover) through time during the period between 12 and 1 ka
for the 36 site-groups (time slices cover entire period) was estimated by detrended
canonical correspondence analysis (DCCA) for each site-group (ter Braak, 1986)
using CANOCO 4.5 (ter Braak and Šmilauer, 2002).

**3. Results**

*Large-scale pattern*

On a glacial-interglacial scale, marked temporal changes in the occurrence and abundance of PFTs are revealed, in particular the high cover of tree PFTs during the Holocene as opposed to the widespread open landscape during the glacial period. In contrast, vegetation changes in northern Asia within the Holocene are rather minor with only slight changes in PFT abundances. Cluster analyses of grouped vegetation records from the Holocene find five clusters (Appendix 6). Their spatial distribution is largely consistent with the distribution of modern vegetation types as characterized by certain PFTs. (1) Records from the forest-steppe ecotone (e.g. G12, G21; Fig. 2A) in north-central China and the Tianshan Mts. (the mentioned geographic locations are indicated in Appendix 7) have high tree PFTs during the middle Holocene. (2) Areas in southern and south-western Siberia and north-eastern China were covered by cool-temperate mixed forest or light taiga with a high diversity of trees throughout the Holocene (e.g. G2, G7, G14, G29; Fig. 2B). (3) The West Siberian Plain and south-eastern Siberia that are presently covered by open dark taiga forests (e.g. G8, G9, G33; Fig. 2C) had an even higher abundance of evergreen conifer trees during the middle Holocene than at present. (4) *Larix* formed light taiga forests in central Yakutia throughout the Holocene (e.g. G25, G26; Fig. 2D). (5) Northern Siberia, which is currently covered by tundra formed by boreal shrubs and herbs, had a higher share of tree PFTs during the middle Holocene (e.g. G28, G39; Fig. 2E).

The turnover in PFT composition is <0.7 SD units in almost all site-groups, except G8 (0.88 SD), G9 (0.73 SD), and G24 (0.76 SD), indicating only slight vegetation change

during the Holocene (Fig. 3). The three site-groups with higher turnover show a
distinct transition from light taiga to dark taiga in the middle Holocene (at ca. 8 ka).
The significant primary vegetation changes (pass the broken-stick test) occur during
different intervals in each site-group. Overall, the middle Holocene (including 8.5, 7.5,
6.5, and 5.5 ka time-slices) has the highest frequency of primary vegetation changes.
Records from inland areas such as the West Siberian Plain, central Yakutia, and
northern Mongolia are characterized by relatively many middle-Holocene splits.
There are seven site-groups whose primary vegetation changes during the early
Holocene (including 11.5, 10.5, and 9.5 ka time-slices), and most of them from the
south-eastern coastal part of the study area. Only three site-groups have late-Holocene
primary vegetation changes (Fig. 3).

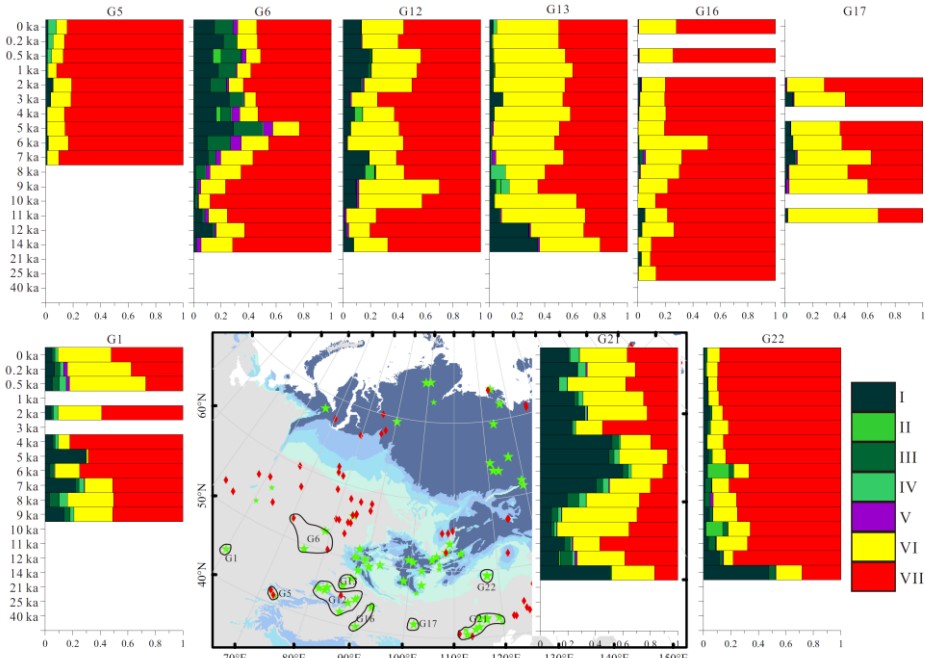

Fig. 2A. Temporal changes of plant functional type (PFT) cover, as proportions, for the site-groups
from the warm temperate forest margin zone. PFT I: evergreen conifer tree; PFT II: deciduous
conifer tree; PFT III: boreal deciduous tree; PFT IV: temperate deciduous tree; PFT V: boreal
shrub; PFT VI: arid-tolerant shrub and herb; and PFT VII: steppe and tundra forb.
*Warm temperate forest margin zone in vicinity of Tianshan Mts. and north-central*
*China (G6, G12, G13, G16, G21, G22)*
Six site-groups from the warm temperate forest-steppe transition zone (G6, G21, G22)

and from the lowlands adjacent to mountainous forest in arid central Asia (G12, G13, G16) are clustered together (Fig. 3). Our results indicate that these areas, which are now dominated by arid-tolerant shrub and steppe species, had more arboreal species, mainly evergreen conifer tree taxa, in the middle Holocene (Fig. 2A). For example, north-central China (G21) has a marked mid-Holocene maximum in forest cover (7–4 ka; mean 51%). However, certain peculiarities are noted: open landscape is reconstructed between 14 and 7 ka in northern Kazakhstan (G6), followed by an abundance of evergreen conifer trees and an increase in boreal deciduous trees that maintain high values (mean 30%) after 7 ka. In the eastern branch of the Tianshan Mts. (G12), evergreen conifer trees are highly abundant from 10 to 7 ka and after 2 ka, while low abundance occurs from 14 to 11 ka and from 6 to 3 ka. In the Gobi desert near the Tianshan Mts. (G16) there was an even higher abundance of arid-tolerant species with no notable temporal trend in abundance of arboreal species. We assume that the high arboreal cover at site-groups G13 and G22 at 14 and 12 ka originates from riverine transport and therefore exclude them from further analyses.

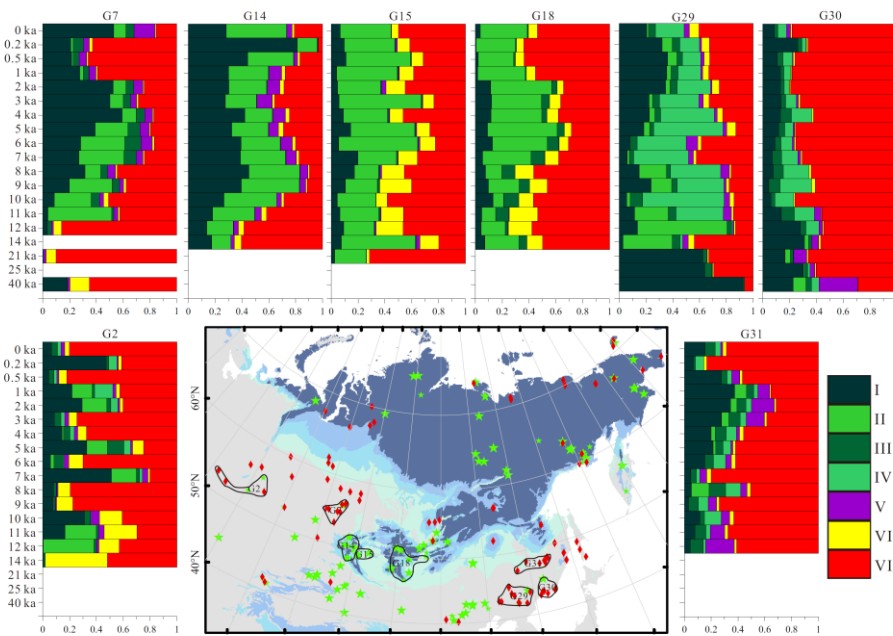

Fig. 2B. Temporal changes of plant functional type (PFT) cover, as proportions, for the site-groups from cool-temperate mixed forest and taiga forest. PFT I: evergreen conifer tree; PFT II: deciduous conifer tree; PFT III: boreal deciduous tree; PFT IV: temperate deciduous tree; PFT V: boreal shrub; PFT VI: arid-tolerant shrub and herb; and PFT VII: steppe and tundra forb.

*Cool-temperate mixed forest and taiga forest in southern and south-western Siberia*
*and north-eastern China (G2, G7, G14, G15, G18, G29, G30, G31)*
Eight site-groups located in (or near) the temperate mixed conifer-deciduous
broadleaved forest zone (G2, G29, G30, G31) and taiga-steppe transition zone (G7,
G14, G15, G18) show similar PFT compositions and temporal evolutions. At these
sites, evergreen conifer tree is the dominant PFT intermixed with other arboreal PFTs,
such as deciduous conifers (*Larix*) in the Altai Mts. and northern Mongolia, and/or
temperate deciduous trees in north-eastern China (Fig. 2B).

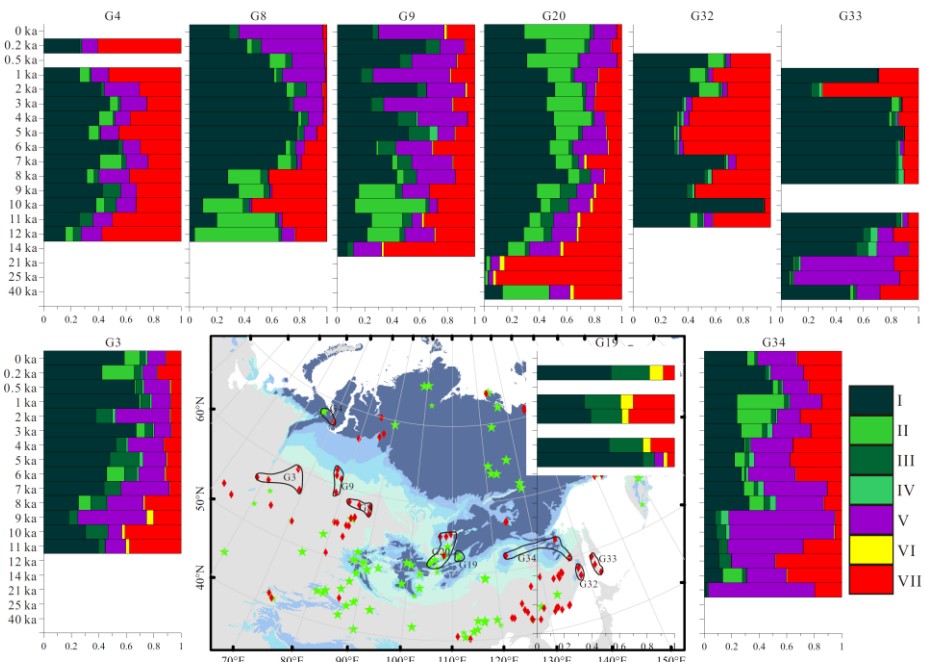


Fig. 2C. Temporal changes of plant functional type (PFT) cover, as proportions, for the site-groups
from dark taiga forest. PFT I: evergreen conifer tree; PFT II: deciduous conifer tree; PFT III:
boreal deciduous tree; PFT IV: temperate deciduous tree; PFT V: boreal shrub; PFT VI:
arid-tolerant shrub and herb; and PFT VII: steppe and tundra forb.
Evergreen conifer tree is the dominant PFT at 40, 25, and 21 ka in the southern part of
north-eastern China (G29), *Larix* then becomes the dominant taxa at 14 and 12 ka,
and temperate deciduous trees increase thereafter and maintain high cover between 11
and 3 ka. After 2 ka, evergreen conifer trees increase to 32% on average while
temperate deciduous trees decrease to 18% on average. While arboreal abundance is
lower in the northern part of north-eastern China (G30, G31) than in the southern part
(G29), it shows a similar temporal pattern (Fig. 2B).

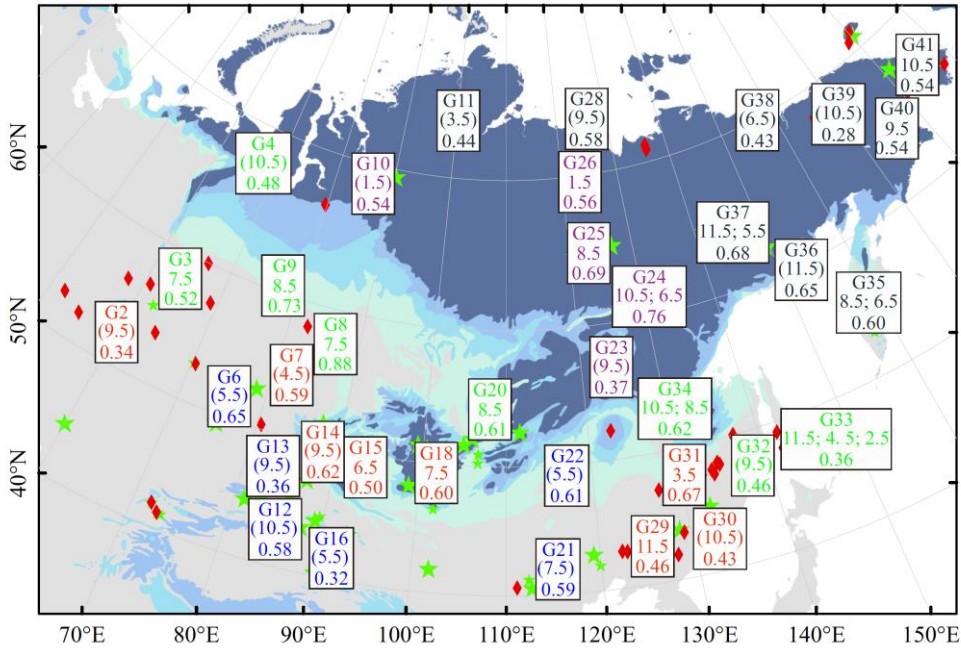


Fig. 3. Clustering results of the 36 site-groups represented by the colour of the boxes, with the age
of primary vegetation changes (middle row of each box; data in brackets means the hierarchical
clustering failed the broken-stick test) and the compositional change (turnover; lower row) during
the Holocene.
Open landscape is revealed for the southern Ural region (G2) with high abundances of
herbaceous species at 14 ka. The cover of *Larix* and evergreen conifer trees increases
after 12 ka and maintains high values thereafter with no notable temporal trend (Fig.
2B).
In the taiga-steppe transition zone, *Larix* is the dominant arboreal taxon, particularly
in the northern Altai Mts. and northern Mongolia (G15, G18). Open landscapes are
inferred at 40, 21, and 12 ka on the southern West Siberian Plain (G7); cover of *Larix*
increases at 11 ka, and evergreen conifer trees increase from 9 ka and become the
dominant forest taxon after 4 ka. The temporal pattern of evergreen conifer trees in
the Altai Mts. (G14) is similar to the southern West Siberian Plain, although *Larix*
maintains high abundances into the late Holocene. Relative to the Altai Mts., the
abundance of evergreen conifer trees for all time windows are lower in the area north
of the Altai Mts. and in northern Mongolia (G15, G18), but their temporal change
patterns are consistent with those of the Altai Mts. (G14; Fig. 2B).
*Dark taiga forest in western and south-eastern Siberia (G3, G4, G8, G9, G20, G32,*
*G33, G34)*
Site-groups with dark taiga forest from western Siberia (G3, G4, G8, G9), the Baikal
region (G20), and south-eastern Siberia (G32, G33, G34) form one cluster sharing
similar PFT compositions dominated by evergreen conifer trees, with *Larix* and boreal
broadleaved shrubs as the common woody taxa during the Holocene (Fig. 2C).
On the West Siberian Plain (G8, G9), high cover of *Larix* is reconstructed during the
early Holocene as well as high woody cover since the middle Holocene formed by
evergreen conifer trees and boreal shrubs. In the Ural region (G3, G4), evergreen
conifer trees dominate the arboreal species throughout the Holocene. The absence of
*Larix* in the early Holocene in this Ural region is a notable difference to the West
Siberian Plain (Fig. 2C).
In the Baikal region (G20), relatively closed landscape is revealed at 40 ka; openness
then increases to >95% at 25 and 21 ka. Since 14 ka, woody cover increases as shown
by a notable rise in evergreen conifer trees from 14 to 8 ka and by increases of *Larix*
after 7 ka (Fig. 2C).
In south-eastern Siberia (G32, G34), arboreal abundance is high in the early and late
Holocene, but low in the middle Holocene. South of Sakhalin Island (G33), closed
landscape is revealed between 40 and 1 ka with >80% woody cover. Evergreen
conifer tree PFT has lower cover than boreal shrub PFT at 25 and 21 ka, but increases
in abundance around 14 ka rising to 83% on average between 11 and 3 ka, and
reduces thereafter (Fig. 2C).
*Light taiga forest in north-western Siberia and central Yakutia (G10, G23, G24, G25,*
*G26)*
Plant composition of this cluster is dominated by *Larix* with high arboreal cover

during the Holocene. Evergreen conifer trees are present at ca. 15% cover between 11 and 2 ka, with high arboreal values (mean 73%) during the Holocene in north-western Siberia (G10). In central Yakutia (G23, G24, G25), evergreen conifer trees increase markedly from ca. 8 ka, 6 ka, and 7 ka, respectively and maintain high cover thereafter, with ca. 60% arboreal cover throughout the Holocene. Evergreen conifer trees are almost absent in the taiga-tundra ecotone (G26; Fig. 2D).

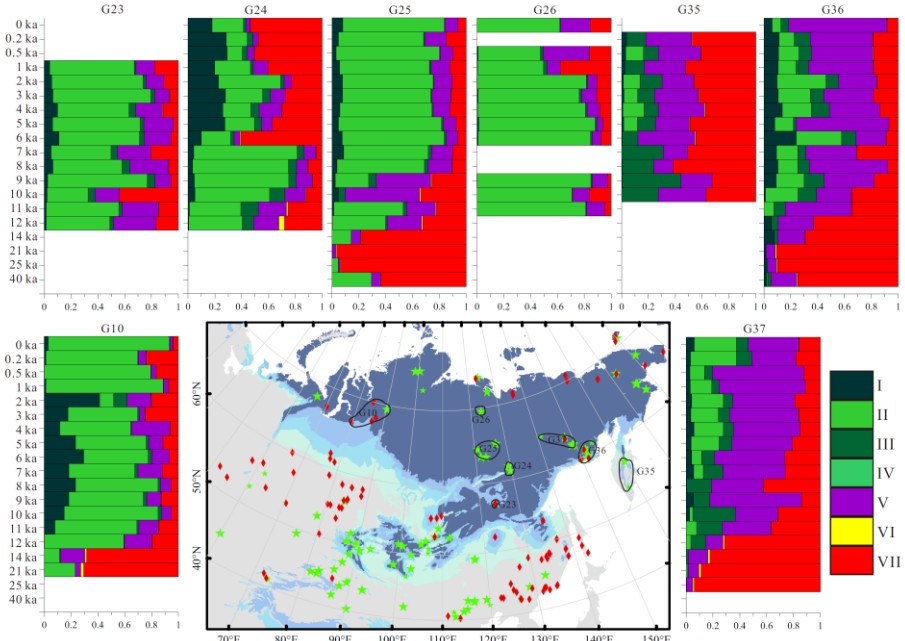

Fig. 2D. Temporal changes of plant functional type (PFT) cover, as proportions, for the site-groups from light taiga forest and taiga-tundra ecotone (G35, G36, G37). PFT I: evergreen conifer tree; PFT II: deciduous conifer tree; PFT III: boreal deciduous tree; PFT IV: temperate deciduous tree; PFT V: boreal shrub; PFT VI: arid-tolerant shrub and herb; and PFT VII: steppe and tundra forb.

*Tundra on the Taymyr Peninsula and taiga-tundra ecotone in north-eastern Siberia (G11, G28, G35, G36, G37, G38, G39, G40, G41)*

Plant compositions of this cluster are characterized by high abundances of boreal shrubs and tundra forbs. *Larix* is the only tree species on the Taymyr Peninsula (G11) and its abundance increases from 18% at 14 ka to 60% at 10 ka, and then decreases to 18% at 5 ka. The landscape of the north Siberian coast (G28) is dominated by shrub tundra from 14 ka to 10 ka, then *Larix* increases sharply and maintains high values between 9 and 6 ka. After 5 ka, *Larix* reduces, and shrub tundra becomes the

dominant landscape again (Fig. 2E).

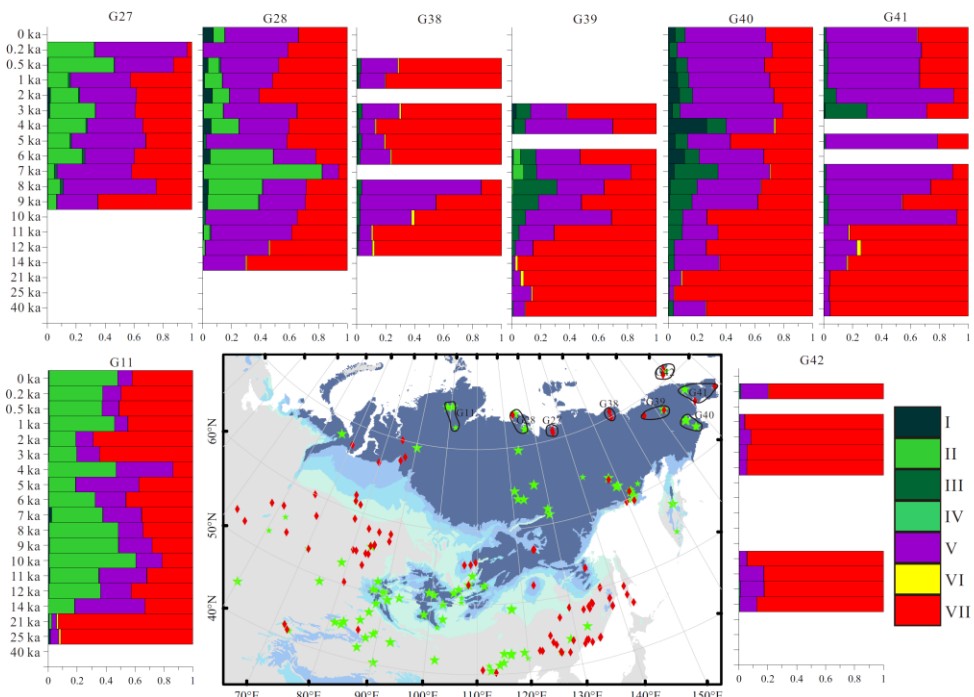

Fig. 2E. Temporal changes of plant functional type (PFT) cover, as proportions, for the site-groups from tundra and taiga-tundra ecotone. PFT I: evergreen conifer tree; PFT II: deciduous conifer tree; PFT III: boreal deciduous tree; PFT IV: temperate deciduous tree; PFT V: boreal shrub; PFT VI: arid-tolerant shrub and herb; and PFT VII: steppe and tundra forb.

In north-eastern Siberia, arboreal cover shows a decreasing trend from southerly site-groups (G35, G36, G37; Fig. 2D) to northerly ones (G40, G38, G39, G41) following the increasing latitude. In the Olsky District, temporal patterns of vegetation changes in G37 are consistent with G36, with stable vegetation during the Holocene and increases in evergreen conifer tree abundance from ca. 9 ka. Arboreal composition on the southern Kamchatka Peninsula (G35) is dominated by boreal deciduous trees during the first stage of the Holocene, followed by rising abundances of *Larix* and evergreen conifer trees from 5 ka.

In north-eastern Siberia (G40, G38, G39, G41), the landscape is dominated by forb tundra with sparse shrubs between 40 and 21 ka; the cover of shrubs increases at 14 ka and arboreal cover (dominated by boreal deciduous trees) increases in the early Holocene (11 or 10 ka). Shrubs maintain a high abundance throughout the Holocene,

while trees peak between 10 and 2 ka generally (Fig. 2E).
**4. Discussion**
*4.1 Land-cover changes and potential biases*
The overall patterns of pollen-based REVEALS estimates of land cover are generally
consistent with previous vegetation reconstructions. Although only a few site-groups
cover the period from 40 to 21 ka, a consistent vegetation signal indicates that
relatively closed landscapes occurred in south-eastern Siberia, north-eastern China,
and the Baikal region (Fig. 2), while most of Siberia was rather open, particularly
around 21 ka (Fig. 2). These findings are consistent with previous pollen-based
(Tarasov et al., 1998, 2000; Bigelow et al., 2003; Binney et al. 2017; Tian et al., 2018)
and model-estimated biome reconstructions (Tian et al., 2018). During the late
Pleistocene (40, 25, 21, 14 ka), steppe PFT abundance was high in central Yakutia and
north-eastern Siberia (e.g. G25, G36, G37, G39, G40, G41), which may reflect the
expansion of tundra-steppe, consistent with results from ancient sediment DNA which
reveal abundant forb species during the period between 46 and 12.5 ka on the Taymyr
Peninsula (Jørgensen et al., 2012). The tundra-steppe was replaced by light taiga in
southern Siberia and by tundra in northern Siberia at the beginning of Holocene or the
last deglaciation, which is consistent with ancient DNA results (forbs-dominated
steppe-tundra; Willerslev et al., 2014).
During the Holocene, reconstructed land cover for each site-group is generally
consistent with their modern vegetation. The slight vegetation changes are represented
by changes in PFT abundances rather than by changes in PFT presence/absence.
Minor changes are also indicated in the cluster analysis, which shows that plant
compositions and their temporal patterns are consistent among the site-groups within
the same modern vegetation zone (Fig. 3). PFT datasets from only 19 site-groups pass
the broken-stick test for clustering analysis, and most of them have only one
significant vegetation change, further supporting the case that only slight changes
occurred during the Holocene in northern Asia. In addition, the low total amount of

PFT change (turnover) over the Holocene for most site-groups supports the view of slight temporal changes in land cover.

Vegetation turnover on the Tibetan Plateau inferred from pollen percentages is documented to overestimate the strength of vegetation changes (Wang and Herzschuh, 2011). This matches with our results. In central Yakutia, the pollen percentage data indicate a strong vegetation change during the middle Holocene, represented by a sharp increase of *Pinus* pollen, but the strength of the vegetation change is overestimated because of the high PPE of *Pinus*. The PPE-corrected arboreal abundances in central Yakutia after ca. 7 ka with ca. 70% *Larix* and ca. 10% *Pinus* are consistent with modern light taiga (Katamura et al., 2009). Furthermore, the absence of *Pinus* macrofossils in central Yakutia throughout the Holocene (Binney et al., 2009) also suggests a restricted distribution of *Pinus,* possibly to sandy places such as river banks (Isaev et al., 2010).

Pollen-based turnover estimates from southern Norway range between 0.84 to 1.3 SD (mean 1.02 SD) for ten Holocene pollen spectra (Birks, 2007), and from northern Europe between 0.01 (recent) to 0.99 (start of the Holocene) SD for three sites (N Sweden, NW and SE Finland) (Marquer et al., 2014). Moreover, the REVEALS-based turnover estimates (0.3–1) for northern Europe are significantly higher than the pollen-based one (0.2–0.8) from 11 ka to 5.5 ka BP. The same is true for all other regions studied by Marquer et al. (2014) in north-western Europe, and the turnover estimates (pollen- and REVEALS-based) are generally higher at lower latitudes from southern Sweden down to Switzerland and eastwards to Britain and Ireland. These European values are higher than our REVEALS-based turnover estimates (from 0.37 to 0.88 SD, mean 0.66 SD; G3, G8, G9, G23, G24, G25, G36, G37) from a similar latitudinal range (Fig. 3). The fewer parameters used in the turnover calculations for northern Asia (PFTs) compared to Europe (pollen taxa) is a potential reason for the lower turnover obtained in this study. In addition, the PPE-based transformation from pollen percentages to plant abundances may reduce the strength of vegetation changes (Wang and Herzschuh, 2011). Aside from the methodological aspects, the lower

turnover in northern Asia may, at least partly, originate from differences in the environmental history between northern Europe compared with northern Asia, that is glaciation followed by postglacial re-vegetation vs. non-glaciated areas with trees in refugia, respectively, and a maritime climate with temperature-limited vegetation distribution vs. a continental climate with temperature- and moisture-limited vegetation.

We consider the REVEALS-based regional vegetation-cover estimations in this study as generally reliable with reasonable standard errors (Appendix 8) thanks to the thorough selection of records with high quality pollen data and reliable chronologies. In addition, the landscape reconstructions are generally consistent with previous syntheses of past vegetation change (e.g. Tian et al., 2018) and known global climate trends (Marcott et al., 2013), plus the clustering results of PFT abundance are consistent with modern spatial vegetation patterns. That said, this study faced two major methodological challenges, discussed below, that may reduce the reliability of the obtained quantitative land-cover reconstructions; 1) the low number of PPEs and their origin and 2) restrictions with respect to the number, distribution, and type of available sites.

(1) Twenty PPE sets were used which mostly originate from Europe and temperate northern China. The available PPEs were estimated from various environmental and ecological settings, which might cause regional differences in each PPE. And, PPEs of different species within one family or genus were included in our mean PPE calculation for the family or genus, ignoring the inter-species differences. Also, some taxa behave few available PPEs with significant differences (such as *Abies*, *Larix*, *Juglans*, Brassicaceae), and their mean PPE could fail to represent their real pollen productivities. These aspects can cause uncertainty in the mean PPE to some extent. However, we believe that the compiled PPE sets can be used to extract major broad-scale and long-term vegetation patterns because the regional differences in the PPE for most taxa are small compared to the large between-taxa differences. The mean

PPEs used in this REVEALS modelling (Table 2) are broadly consistent with
those obtained from Europe (Mazier et al., 2012). In addition, although there
are no PPEs for the core from the Siberia taiga forest, available studies on
modern pollen composition support the weightings in the applied PPEs for
major taxa in terms of pollen under- or over-representation of vegetation
abundance. For example, modern pollen investigations in north-eastern Siberia
revealed that pollen records from northern *Larix* forest often have less than
13% *Larix* pollen, confirming the low pollen productivity of *Larix* relative to
over-represented pollen taxa such as *Betula* and *Alnus* (Pisaric et al., 2001,
Klemm et al., 2016). Similarly, a study on modern pollen in southern Siberia
(transitional area of steppe and taiga) finds that *Artemisia*, *Betula,* and *Pinus*
are high pollen producers compared to *Larix* (Pelánková et al., 2008). Also,
despite *Larix* being the most common tree in taiga forest in north-central
Mongolia, the pollen abundance of *Larix* is generally lower than 3% (Ma et al.,
2008), implying its low pollen productivity.
(2) In this study, we attempt to reconstruct past landscape changes at a regional
scale. Pollen signals from large lakes are assumed to reflect regional
vegetation patterns (e.g. Sugita et al., 2010; Trondman et al., 2015). If large
lakes are absent in a region, multiple small-sized sites can be used, although
error estimates are usually large (Sugita, 2007; Mazier et al., 2012; Trondman
et al., 2016). In our study, 70% of the time slices for the 42 site-groups include
pollen data from large lakes (i.e. radii >390m), which supports the reliability
of REVEALS reconstructions (Appendix 5). However, sites are unevenly
distributed and occasionally sites from different areas were combined into one
group (G2, G6, G34), which might produce a different vegetation-change
signal because of the broad distribution of these sites (Fig. 1). In addition, the
linear interpolation of pollen abundances for time windows with few pollen
data might be another source of uncertainty, particularly for the late
Pleistocene and its broad time windows (Table 1). Finally, pollen signals from

certain sites and during certain periods may be of water-runoff origin rather

than aerial origin violating the assumption of the REVEALS-model that pollen

is transported by wind.

### *4.2 Driving factors of vegetation changes*

On a glacial-interglacial scale, pollen-based reconstructed land-cover changes in northern Asia are generally consistent with the global climate signal (e.g. sea-surface temperature: Pailler and Bard, 2002; ice-core: Andersen et al., 2004; solar insolation: Laskar et al., 2004; cave deposits: Cheng et al., 2016; Appendix 9). For example, the relatively high arboreal cover at 40 ka (e.g. G20) corresponds with the warm MIS3 record from the Baikal region (Swann et al., 2005). The open landscape at 25 ka and 21 ka (e.g. G25, G36) reflects the cold and dry last glacial maximum (e.g. Swann et al., 2010). Furthermore, the relatively high arboreal cover during the Holocene is consistent with the warm and wet climate (occurring in most site-groups). The primary vegetation change in north-eastern China (G29, G30) occurs in the early Holocene (11.5 and 10.5 ka), caused by the rapid increase in abundance of temperate deciduous trees, which may reflect the warmer climate and enhanced summer monsoon known from that region at the beginning of the Holocene (Hong et al., 2009, Liu et al., 2014).

A sensitivity analysis of model-based biome estimation reveals that precipitation plays an important or even dominant role in controlling vegetation changes in arid central Asia (e.g. Tian et al., 2018). The climate of central Asia during the early Holocene is inferred to be quite dry and moisture increase occurs at ca. 8 ka revealed by a series of multi-proxy syntheses (Chen et al., 2008, 2016; Xie et al., 2018) and model-based estimations (Jin et al., 2012). In the taiga-steppe transition zone (south-eastern Siberia and north-central Asia; e.g. G6, G12, G14, G18), relatively open landscape is reconstructed for the early Holocene and abundances of forest taxa increase after ca. 8 ka, which are consistent with the moisture evolution, and imply the importance of moisture in controlling vegetation changes. Our results support the prediction of an expansion of steppe in the present forest–steppe ecotone of southern Siberia in

response to a warmer and drier climate in the future (Tchebakova et al., 2009). During
the late Holocene, the decreases in forest cover in the forest–steppe ecotone of
north-central China and central Asia are ascribed to the drying or cooling climate
respectively by sensitivity analysis (Tian et al., 2018). Previous studies argued that the
enhanced human impacts might be important factor for the reduce in forest cover (e.g.
Ren, 2017), however our study fails to determine   its contribution on vegetation
changes.
High abundances of *Larix* or boreal deciduous woody taxa (mostly shrubs) pollen
occur in northern Siberia (e.g. G28, G38, G39, G40) during the middle Holocene,
which is now covered by tundra. This is consistent with non-vegetation climate
records of a mid-Holocene temperature maximum (e.g. Biskaborn et al, 2012;
Nazarova et al., 2013). This result indicates that the boreal treeline in northern Siberia
reacts sensitively to warming on millennial time-scales, which contrasts with the
observed lack of response on a decadal time-scale (Wieczoreck et al., 2017). This may
point to a highly non-linear vegetation–climate relationship in northern Siberia.
Our results indicate that climate change is the major factor driving land-cover change
in northern Asia on a long temporal scale. However, climate change cannot fully
explain the changes in arboreal taxa abundance for the West Siberian Plain (G8, G9)
and sandy places in central Yakutia (G23, G24, G25). In addition to climate, changes
in permafrost condition (Vandenberghe et al., 2014) and fire regime may have played
a central role in vegetation change. *Larix* is the dominant arboreal taxon during the
early Holocene (ca. between 12 and 8 ka), which is replaced by evergreen conifer
trees, mostly pine and spruce at 8 or 7 ka. *Larix* can survive on permafrost with an
active-layer depth of <40 cm (Osawa et al., 2010) and a high fire frequency, while
pine trees can only grow on soil with >1.5m active-layer (Tzedakis and Bennett, 1995)
and spruce is a fire-avoider. Probably the compositional change of boreal trees was
not in equilibrium with climate but rather driven by changes in the permafrost and fire
characteristics that were themselves affected by forest composition, resulting in
complex feedbacks. This explanation would be in agreement with the finding of
Herzschuh et al. (2016) that the boreal forest composition of nearby refugia during a
glacial influences the initial interglacial forest composition that is then only slowly
replaced by a forest composition that is in equilibrium with climate.
Population changes of herbivores could also be an important factor for vegetation
change at a regional scale during certain intervals (Zimov et al., 1995; Guthrie, 2006).
As with our pollen-based land-cover reconstruction, a circumpolar ancient DNA
metabarcoding study confirms the replacement of steppe-like tundra by moist tundra
with abundant woody plants at the Pleistocene–Holocene transition (Willerslev et al.,
2014). According to Zimov et al. (1995, 2012), such a change cannot be explained by
climate change alone, and thus a reduced density of herbivores is considered to be a
major driving factor of steppe composition reduction, since a reduced number of
herbivores is insufficient to maintain the open steppe landscapes and so causes a
decrease in steppe area (Zimov et al., 1995; Guthrie, 2006). Our land-cover
reconstruction fails to address the contribution of herbivores to vegetation changes,
but the extinction of herbivorous megafauna would add to the complexity of the
interactions among vegetation, climate and permafrost.
**5. Conclusions**
Regional vegetation based on pollen data has been estimated using the REVEALS
model for northern Asia during the last 40 ka. Relatively closed land cover was
replaced by open landscapes in northern Asia during the transition from MIS 3 to the
last glacial maximum. Abundances of woody components increase again from the last
deglaciation or early Holocene. Pollen-based REVEALS estimates of plant
abundances should be a more reliable reflection of the vegetation as pollen may
overestimate the turnover, and indicates that the vegetation was quite stable during the
Holocene as only slight changes in the abundances of PFTs were recorded rather than
mass expansion of new PFTs. From comparisons of our results with other data, we
infer that climate change is likely the primary driving factor for vegetation changes on
a glacial-interglacial scale. However, the extension of evergreen conifer trees since ca.

8–7 ka throughout Siberia could reflect vegetation-climate disequilibrium at a long-term scale caused by the interaction of climate, vegetation, fire, and permafrost, which could be a palaeo-analogue not only for the recent complex vegetation response to climate changes but also for the vegetation prediction in future.

**Data availability**. The used fossil pollen dataset with the re-established age-depth model for each pollen record have been made publicly available in Pangaea (https://doi.pangaea.de/10.1594/PANGAEA.898616).

**Acknowledgements.** The authors would like to express their gratitude to all the palynologists who, either directly or indirectly, contributed their pollen records and PPE results to our study. This research was supported by the German Research Foundation (DFG) and PalMod project (BMBF). FL and MJG thank the Faculty of Health and Life Science of Linnaeus University (Kalmar, Sweden), the China-Swedish STINT Exchange Grant 2016-2018 and the Swedish Strategic Research Area on ModElling the Regional and Global Earth system (MERGE) for financial support. This study is a contribution to the Past Global Changes (PAGES) LandCover6k working group project.

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

Zudin, A.N., and Votakh, M.R.: The stratigraphy of Pliocene and Quaternary strata of
Priobskogo Plateau, Nauka, Novosibirsk, 1977 (in Russian).

Appendices
**Appendix 1** Distribution of the 203 fossil pollen sites together with the modern permafrost extent in northern Asia. The number of each site is used as its site ID in
Appendix 2.

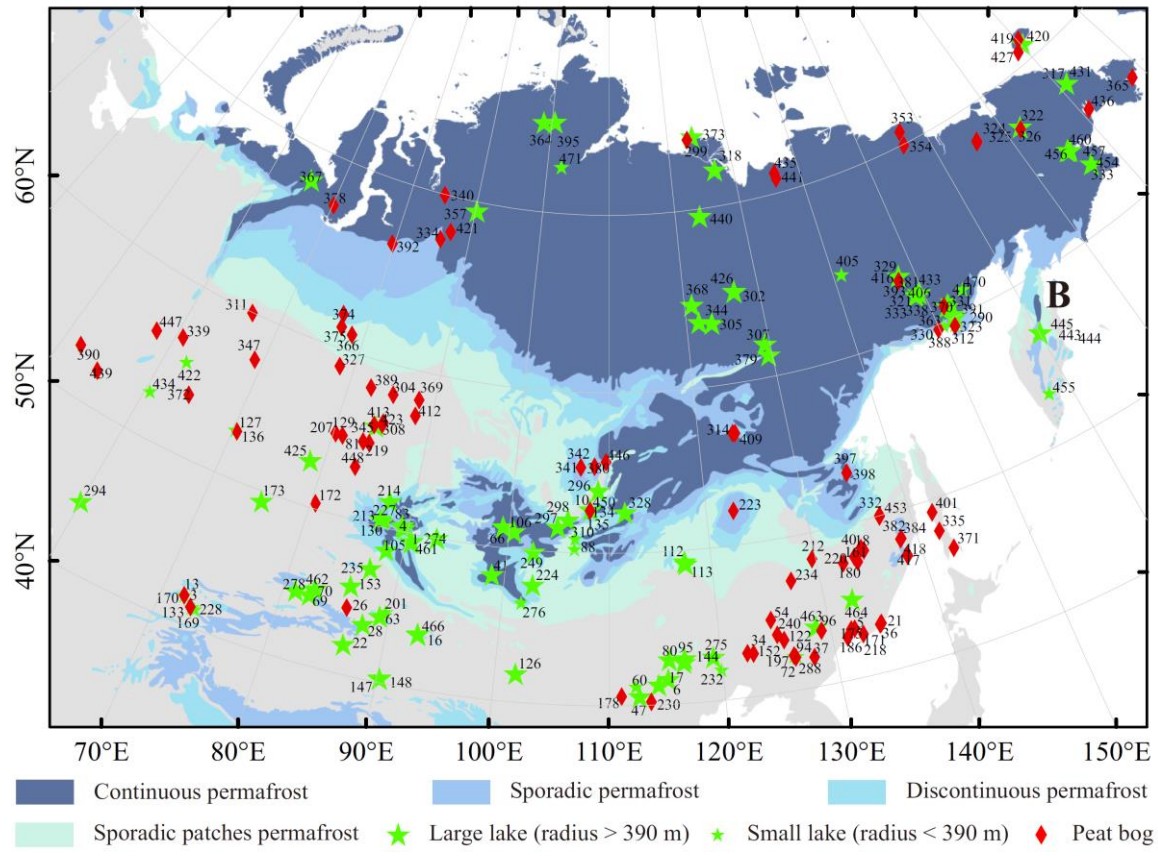


 **Appendix 2** Metadata for all pollen records used in this study. Original publications list see https://doi.pangaea.de/10.1594/PANGAEA.898616.

| Group | Site ID | Site | Lat. | Long. | Elev. (m) | Basin type | Pollen count | Area (ha) | Radius (m) | Dating method | Num. of dating | Time span (cal ka BP) | Resol. (year) | Reference |
|---|---|---|---|---|---|---|---|---|---|---|---|---|---|---|
| G1 | 294 | Aral Lake | 44.42 | 59.98 | 53 | Lake | Yes | 330000 | 32410 | $^{14}$C | 4U | 8.7-0 | 260 | Aleshinskaya, Z.V. unpublished. |
| G2 | 372 | Mokhovoye | 53.77 | 64.25 | 178 | Bog | Yes | 20 | 252 | $^{14}$C | 4C+1E | 6.0-0 | 180 | Kremenetskii et al., 1994 |
| G2 | 439 | Novienky peat bog | 52.24 | 54.75 | 197 | Bog | Yes | - | - | $^{14}$C | 1U | 4.5-0 | 270 | López-García et al., 2003 |
| G2 | 422 | Zaboinoe Lake | 55.53 | 62.37 | 275 | Lake | Yes | 6 | 138 | $^{14}$C | 1U | 12.3-0.1 | 220 | Khomutova and Pushenko, 1995 |
| G2 | 434 | Lake Fernsehsee | 52.83 | 60.50 | 290 | Lake | Yes | 0 | 38 | $^{14}$C | 10A | 9.1-0.4 | 220 | Stobbe et al., 2015 |
| G2 | 390 | Pobochnoye | 53.03 | 51.84 | 81 | Bog | No | 79 | 500 | $^{14}$C | 10C+6E | 14.4-0 | 540 | Kremenetski et al., 1999 |
| G3 | 311 | Chesnok Peat | 60.00 | 66.50 | 42 | Bog | Yes | - | - | $^{14}$C | 7C | 10.6-0.5 | 280 | Volkova, 1966 |
| G3 | 347 | Komaritsa Peat | 57.50 | 69.00 | 42 | Bog | Yes | - | - | $^{14}$C | 10C | 10.5-0.5 | 350 | Volkova, 1966 |
| G3 | 447 | UstMashevskoe | 56.32 | 57.88 | 220 | Bog | Yes | 30 | 309 | $^{14}$C | 5C | 7.8-0 | 150 | Panova et al., 1996 |
| G3 | 339 | Karasieozerskoe | 56.77 | 60.75 | 230 | Bog | Yes | 914 | 1706 | $^{14}$C | 3A | 5.9-0.1 | 190 | Panova, 1997 |
| G4 | 378 | Nulsaveito | 67.53 | 70.17 | 57 | Bog | Yes | - | - | $^{14}$C | 4A+1C | 8.4-6.4 | 70 | Panova, 1990 |
| G4 | 367 | Lyadhej-To Lake | 68.25 | 65.75 | 150 | Lake | Yes | 197 | 792 | $^{14}$C | 14A+6E | 12.5-0.3 | 170 | Andreev et al., 2005 |
| G5 | 169 | Nizhnee Lake | 41.30 | 72.95 | 1371 | Lake | No | - | 70 | $^{14}$C | 4E | 1.5-0 | 100 | Beer et al., 2008 |
| G5 | 228 | Verkhnee Lake | 41.30 | 72.95 | 1440 | Lake | No | 1 | 60 | $^{14}$C | 5E | 1.5-0 | 100 | Beer et al., 2008 |
| G5 | 3 | Ak Terk Lake | 41.28 | 72.83 | 1748 | Bog | No | - | - | $^{14}$C | 2A | 7.5-0 | 200 | Beer et al., 2008 |
| G5 | 133 | Kosh Sas | 41.85 | 71.97 | 1786 | Bog | No | - | - | $^{14}$C | 1A | 3.5-0 | 100 | Beer et al., 2008 |
| G5 | 170 | Ortok Lake | 41.23 | 73.25 | 1786 | Lake | No | - | 60 | $^{14}$C | 5A | 1-0 | 100 | Beer et al., 2008 |
| G5 | 13 | Bakaly Lake | 41.87 | 71.97 | 1879 | Lake | No | 1 | 50 | $^{14}$C | 4A | 7-0 | 195 | Beer et al., 2008 |
| G6 | 425 | Big Yarovoe Lake | 52.85 | 78.63 | 79 | Lake | Yes | 6362 | 4500 | inclination | - | 4.3-0 | 190 | Rudaya et al., 2012 |

| | | | | | | | | | | | | | |
|---|---|---|---|---|---|---|---|---|---|---|---|---|---|
| G6 | 172 | Ozerki | 50.40 | 80.47 | 210 | Bog | Yes | - | - | [14]C | 3A+13C | 14.5-0 | 300 | Tarasov et al., 1997 |
| G6 | 127 | Karas'e Lake | 53.03 | 70.22 | 435 | Lake | Yes | 17 | 235 | [14]C | 6U | 5.5-0 | 170 | Tarasov and Kremenetskii. 1995 |
| G6 | 136 | Kotyrkol | 52.97 | 70.42 | 439 | Bog | Yes | - | - | [14]C | 8U | 4.5-0.5 | 180 | Tarasov and Kremenetskii. 1995 |
| G6 | 173 | Pashennoe Lake | 49.37 | 75.40 | 871 | Lake | Yes | 64 | 451 | [14]C | 5D+5E | 9.5-0 | 280 | Tarasov and Kremenetskii. 1995 |
| G7 | 81 | Gladkoye Bog | 55.00 | 83.33 | 80 | Bog | Yes | - | - | [14]C | 13C | 11-0.5 | 170 | Firsov et al., 1982 |
| G7 | 308 | Chaginskoe Mire | 56.45 | 84.88 | 80 | Bog | Yes | 10 | 175 | [14]C | 2C | 8.8-0 | 320 | Blyakharchuk, 2003. |
| G7 | 345 | Kirek Lake | 56.10 | 84.22 | 90 | Lake | Yes | 52 | 407 | [14]C | 3G | 10.5-1.5 | 190 | Blyakharchuk, 2003 |
| G7 | 413 | Tom' River Peat | 56.17 | 84.00 | 100 | Bog | Yes | - | - | [14]C | 6C | 10.1-0.2 | 390 | Arkhipov and Votakh, 1980 |
| G7 | 423 | Zhukovskoye mire | 56.33 | 84.83 | 106 | Bog | Yes | - | - | [14]C | 9C+6H | 11.2-0 | 130 | Borisova et al., 2011 |
| G7 | 219 | Tolmachevsko | 55.00 | 84.00 | 110 | Bog | Yes | - | - | [14]C | 1A+3C | 13-1.5 | 400 | Volkov and Arkhipov, 1978 |
| G7 | 207 | Suminskoye | 55.00 | 80.25 | 135 | Bog | Yes | - | - | [14]C | 8A | 3-0 | 200 | Klimanov, 1976 |
| G7 | 129 | Kayakskoye | 55.00 | 81.00 | 150 | Bog | Yes | - | - | [14]C | 5C | 6.5-0 | 210 | Levina et al., 1987 |
| G7 | 448 | Kalistratikha | 53.33 | 83.25 | 190 | Bog | Yes | - | - | [14]C | 4A | 39.0-12.7 | 1870 | Zudin and Votakh, 1977 |
| G8 | 389 | Petropavlovka | 58.33 | 82.50 | 100 | Bog | Yes | - | - | [14]C | 4C+1E | 10.5-0.1 | 160 | Blyakharchuk, 1989 |
| G8 | 304 | Bugristoe | 58.25 | 85.17 | 130 | Bog | Yes | - | - | LSC | 4C+1E | 11.5-5.0 | 100 | Blyakharchuk, 1989 |
| G8 | 369 | Maksimkin Yar | 58.33 | 88.17 | 150 | Bog | Yes | - | - | [14]C | 4C | 8.3-0.2 | 170 | Blyakharchuk, 1989 |
| G8 | 412 | Teguldet | 57.33 | 88.17 | 150 | Bog | Yes | - | - | LSC | 3C | 7.3-2.4 | 90 | Blyakharchuk, 1989 |
| G9 | 374 | Nizhnevartovsk | 62.00 | 76.67 | 54 | Bog | Yes | - | - | [14]C | 3A+7C | 11.1-0 | 300 | Neustadt and Zelikson, |

1985

| | | | | | | | | | | | | | | |
|---|---|---|---|---|---|---|---|---|---|---|---|---|---|---|
| G9 | 375 | Nizhnevartovskoye | 61.25 | 77.00 | 55 | Bog | Yes | - | - | ¹⁴C | 1A+12C+1E | 12.6-0 | 380 | Neishtadt, 1976 |
| G9 | 327 | Entarnoye Peat | 59.00 | 78.33 | 65 | Bog | Yes | - | - | ¹⁴C | 5C | 14.9-0.9 | 460 | Neishtadt, 1976 |
| G9 | 366 | Lukaschin Yar | 61.00 | 78.50 | 65 | Bog | Yes | - | - | ¹⁴C | 13C | 10.9-0.3 | 430 | Neishtadt, 1976 |
| G10 | 334 | Igarka Peat | 67.67 | 86.00 | 45 | Bog | Yes | 244 | 881 | ¹⁴C | 1A+2C | 10.9-5.9 | 230 | Kats, 1953 |
| G10 | 392 | Pur-Taz Peatland | 66.70 | 79.73 | 50 | Bog | Yes | 5 | 126 | ¹⁴C | 5A | 10.3-4.7 | 80 | Peteet et al., 1998 |
| G10 | 340 | Karginskii Cape | 70.00 | 85.00 | 60 | Bog | Yes | - | - | ¹⁴C | 13C | 8.9-3.5 | 290 | Firsov et al., 1972 |
| G10 | 421 | Yenisei | 68.17 | 87.15 | 68 | Bog | No | - | - | ¹⁴C | 7C | 6.5-1.6 | 110 | Andreev and Klimanov 2000 |
| G10 | 357 | Lake Lama | 69.53 | 90.20 | 77 | Lake | Yes | 64245 | 14300 | ¹⁴C | 26A+4D+4E | 19.5-0 | 170 | Andreev et al., 2004 |
| G11 | 471 | 11-CH-12A Lake | 72.40 | 102.29 | 60 | Lake | Yes | 3 | 100 | ¹⁴C+Pb/Cs | 8A+7E | 7.0-0.1 | 110 | Klemm et al., 2015 |
| G11 | 364 | Levinson-Lessing Lake | 74.47 | 98.64 | 26 | Lake | Yes | 2145 | 2613 | ¹⁴C | 29A+1B+19E | 35.3-0 | 390 | Andreev et al., 2003 |
| G11 | 395 | SAO1 | 74.55 | 100.53 | 32 | Lake | Yes | 456000 | 38098 | ¹⁴C | 6A+5C | 57.9-0 | 1320 | Andreev et al., 2003 |
| G12 | 462 | Aibi Lake | 45.02 | 82.83 | 200 | Lake | Yes | 100885 | 17920 | ¹⁴C | 8E | 12.6-0 | 65 | Wang et al., 2013 |
| G12 | 69 | Ebinur Lake | 44.55 | 82.45 | 212 | Lake | Yes | 46421 | 12156 | ¹⁴C | 7U | 13-0 | 900 | Wen and Qiao, 1990 |
| G12 | 70 | Ebinur Lake_SW | 45.00 | 82.80 | 212 | Lake | Yes | 46421 | 12156 | ¹⁴C | 6U | 8.5-1.5 | 780 | Lin, 1994 |
| G12 | 26 | Caotanhu Lake | 44.42 | 86.02 | 380 | Bog | Yes | 2760 | 2964 | ¹⁴C | 5C | 8.5-0 | 150 | Zhang Y. et al., 2008 |
| G12 | 63 | Dongdaohaizi Lake | 44.70 | 89.56 | 430 | Lake | Yes | 20 | 252 | ¹⁴C | 8U | 5.5-0 | 85 | Yan et al., 2004 |
| G12 | 201 | Sichanghu Lake | 44.31 | 89.14 | 589 | Lake | Yes | 2000 | 2523 | ¹⁴C | 4U | 1-0 | 50 | Zhang Y. et al., 2004b |
| G12 | 22 | Bosten Lake | 41.97 | 86.55 | 1050 | Lake | No | 96608 | 17536 | ¹⁴C | 5U | 13-0 | 420 | Xu, 1998 |
| G12 | 28 | Chaiwopu Lake | 43.55 | 87.78 | 1100 | Lake | No | 3101 | 3142 | ¹⁴C | 2U | 10-0 | 845 | Li and Yan, 1990 |
| G12 | 278 | Sayram Lake | 44.57 | 81.15 | 2072 | Lake | Yes | 45800 | 12074 | ¹⁴C | 12E | 13.8-0.1 | 90 | Jiang et al., 2013 |
| G13 | 153 | Manas Lake | 45.83 | 85.92 | 251 | Lake | Yes | 55000 | 13231 | ¹⁴C | 7C | 13.5-1 | 210 | Sun et al., 1994 |
| G13 | 235 | Wulungu Lake | 47.22 | 87.30 | 479 | Lake | Yes | 67019 | 430 | ¹⁴C+Pb/Cs | 1C | 9-0 | 80 | Liu X.Q. et al., 2008 |

| | | | | | | | | | | | | | | |
|---|---|---|---|---|---|---|---|---|---|---|---|---|---|---|
| G14 | 214 | Teletskoye Lake | 51.72 | 87.65 | 1900 | Lake | Yes | 16610 | 7271 | [14]C+Pb/Cs | 6E | 1-0 | 20 | Andreev et al., 2007 |
| G14 | 227 | Uzunkol Lake | 50.48 | 87.11 | 1985 | Lake | No | 123 | 625 | [14]C | 2A | 17.5-0 | 210 | Blyakharchuk et al., 2004 |
| G14 | 130 | Kendegelukol Lake | 50.51 | 87.64 | 2050 | Lake | No | 5 | 130 | [14]C | 7E | 16-1 | 260 | Blyakharchuk et al., 2004 |
| G14 | 105 | Hoton Nur Lake | 48.62 | 88.35 | 2083 | Lake | Yes | 5021 | 3998 | [14]C | 4A | 6-0 | 60 | Rudaya et al., 2009 |
| G14 | 213 | Tashkol Lake | 50.45 | 87.67 | 2150 | Lake | No | - | 150 | [14]C | 3C | 16-3 | 250 | Blyakharchuk et al., 2004 |
| G14 | 4 | Akkol Lake | 50.25 | 89.63 | 2204 | Lake | No | 388 | 1111 | [14]C | 12E | 13.5-0 | 250 | Blyakharchuk et al., 2007 |
| G14 | 83 | Grusha Lake | 50.38 | 89.42 | 2413 | Lake | No | 130 | 644 | [14]C | 3A+13E | 14-1.5 | 250 | Blyakharchuk et al., 2007 |
| G15 | 274 | Bayan Nuur | 50.00 | 93.00 | 932 | Lake | No | 2968 | 3073 | [14]C | 7E | 15.7-0.2 | 210 | Krengel, 2000 |
| G15 | 1 | Achit Nur Lake | 49.50 | 90.60 | 1435 | Lake | No | 29700 | 9723 | [14]C | 4E | 14-0.5 | 700 | Gunin et al., 1999 |
| G15 | 461 | Achit Nuur | 49.42 | 90.52 | 1444 | Lake | No | 29700 | 9723 | [14]C | 10E | 20.2-0 | 250 | Sun et al., 2013 |
| G16 | 148 | Lop Nur_1998 | 40.28 | 90.25 | 780 | Lake | No | 535000 | 41267 | [14]C | 3U | 22-2 | 2000 | Yan et al., 1998 |
| G16 | 147 | Lop Nur_1983 | 40.33 | 90.25 | 800 | Lake | Yes | 535000 | 41267 | [14]C | 3U | 22-0.5 | 1600 | Yan et al., 1983 |
| G16 | 16 | Barkol Lake | 43.62 | 92.80 | 1575 | Lake | Yes | 11300 | 5997 | [14]C | 1A+10E | 10-0 | 115 | Tao et al., 2009 |
| G16 | 466 | Balikun Lake | 43.68 | 92.80 | 1575 | Lake | Yes | 7897 | 5014 | [14]C | 1D+5E | 30.5-9 | 250 | An et al., 2013 |
| G17 | 126 | Juyan Lake | 41.89 | 101.85 | 892 | Lake | Yes | 72000 | 15139 | [14]C | 5E | 10.5-1.5 | 140 | Herzschuh et al., 2004 |
| G18 | 88 | Gun Nur Lake | 50.25 | 106.60 | 600 | Lake | No | 33 | 325 | [14]C | 7E | 11-0 | 320 | Gunin et al., 1999 |
| G18 | 249 | Yamant Nur Lake | 49.90 | 102.60 | 1000 | Lake | No | 58 | 430 | [14]C | 4E | 15.5-0.5 | 360 | Gunin et al., 1999 |
| G18 | 224 | Ugii Nuur Lake | 47.77 | 102.77 | 1330 | Lake | No | 2456 | 2796 | [14]C | 2C | 9-0 | 85 | Wang et al., 2011 |
| G18 | 66 | Dood Nur Lake | 51.33 | 99.38 | 1538 | Lake | No | 6400 | 4514 | [14]C | 2E | 14-0 | 740 | Gunin et al., 1999 |
| G18 | 106 | Hovsgol Lake | 51.10 | 100.50 | 1645 | Lake | Yes | 276000 | 29640 | [14]C | 5E | 12-2.5 | 190 | Prokopenko et al., 2007 |
| G18 | 276 | Khuisiin Lake | 46.60 | 101.80 | 2270 | Lake | Yes | 4 | 118 | [14]C+Pb/Cs | 6E | 1.2-0 | 17 | Tian et al., 2013 |
| G18 | 41 | Daba Nur Lake | 48.20 | 98.79 | 2465 | Lake | No | 157 | 707 | [14]C | 5E | 13-0 | 520 | Gunin et al., 1999 |
| G19 | 328 | Bolshoe Eravnoe Lake | 52.58 | 111.67 | 947 | Lake | Yes | 9503 | 5500 | [14]C | 3E | 7.3-0.2 | 710 | Vipper, 2010 |
| G20 | 10 | Baikel Lake | 52.08 | 105.87 | 130 | Lake | No | 3150000 | 100134 | [14]C | 12A | 22-0 | 370 | Demske et al., 2005 |

| | | | | | | | | | | | | | |
|---|---|---|---|---|---|---|---|---|---|---|---|---|---|
| G20 | 296 | Baikal Lake-CON01-603-5 | 53.95 | 108.91 | 446 | Lake | Yes | 3150000 | 100134 | [14]C | 10D | 15.8-0 | 270 | Demske et al., 2005 |
| G20 | 135 | Lake Kotokel_2010 | 52.78 | 108.12 | 458 | Lake | Yes | 6900 | 4687 | [14]C | 11E | 47-0 | 220 | Bezrukova et al., 2010 |
| G20 | 134 | Lake Kotokel_2009 | 52.78 | 108.12 | 458 | Lake | Yes | 6900 | 4687 | [14]C | 3E | 15-0 | 500 | Tarasov et al., 2009 |
| G20 | 310 | Chernoe Lake | 50.95 | 106.63 | 500 | Lake | Yes | - | 250 | [14]C | 4E | 7-0.7 | 620 | Vipper, 2010 |
| G20 | 297 | Baikal Lake-CON01-605-3 | 51.59 | 104.85 | 675 | Lake | Yes | 3150000 | 100134 | [14]C | 5D | 17.7-0 | 200 | Demske et al., 2005 |
| G20 | 380 | Okunayka | 55.52 | 108.47 | 802 | Bog | Yes | - | - | [14]C | 6C | 8.3-2.0 | 120 | Bezrukova et al., 2011 |
| G20 | 446 | Ukta Creek mouth | 55.80 | 109.70 | 906 | Bog | Yes | - | - | [14]C | 3U | 5.1-0 | 160 | Bezrukova et al., 2006 |
| G20 | 450 | Cheremushka Bog | 52.75 | 108.08 | 1500 | Bog | Yes | - | - | [14]C | 6C | 33.5-0 | 460 | Shichi et al., 2009 |
| G20 | 298 | Baikal Lake-CON01-605-5 | 51.58 | 104.85 | 492 | Lake | Yes | 3150000 | 100134 | [14]C | 12D | 11.5-0 | 130 | Demske et al., 2005 |
| G20 | 341 | Khanda-1 | 55.44 | 107.00 | 867 | Bog | Yes | - | - | [14]C | 3C | 3.1-0.3 | 50 | Bezrukova et al., 2011 |
| G20 | 342 | Khanda | 55.44 | 107.00 | 867 | Bog | Yes | - | - | [14]C | 6C | 5.8-0 | 140 | Bezrukova et al., 2011 |
| G21 | 275 | Qiganhu Lake | 42.90 | 119.30 | 600 | Lake | Yes | 190 | 778 | [14]C | 5E | 12.1-6.7 | 35 | Hu et al., 2016 |
| G21 | 232 | Wangyanggou | 42.07 | 119.92 | 751 | Lake | No | 13 | 200 | [14]C | 1A+3E | 5-0 | 85 | Li et al., 2006 |
| G21 | 230 | Wangguantun | 40.27 | 113.67 | 800 | Bog | Yes | - | - | [14]C | 1A+4F | 8-3 | 310 | Kong and Du, 1996 |
| G21 | 6 | Anguli Nur Lake | 41.33 | 114.37 | 1000 | Lake | Yes | 4264 | 3684 | [14]C | 2U | 14-10.5 | 520 | Li et al., 1990 |
| G21 | 178 | Qasq | 40.67 | 111.13 | 1000 | Bog | Yes | - | - | [14]C | 2E | 10-0 | 90 | Wang et al., 1997 |
| G21 | 47 | Daihai Lake_2004 | 40.58 | 112.67 | 1220 | Lake | Yes | 16000 | 7136 | [14]C | 8E | 11.5-0 | 215 | Xiao et al., 2004 |
| G21 | 80 | Gaoximage Lake | 42.95 | 115.37 | 1253 | Lake | No | 100000 | 17841 | [14]C | 4E | 6-0 | 150 | Li C.Y. et al., 2003 |
| G21 | 95 | Haoluku Lake | 42.96 | 116.76 | 1295 | Lake | No | 1384 | 2099 | [14]C | 4E | 11.5-0 | 250 | Wang et al., 2001 |
| G21 | 17 | Bayanchagan Lake | 41.65 | 115.21 | 1355 | Lake | Yes | 636 | 1423 | [14]C | 2B+7E | 11.5-0 | 250 | Jiang et al., 2006 |
| G21 | 144 | Liuzhouwan Lake | 42.71 | 116.68 | 1365 | Lake | No | 288 | 957 | [14]C | 3E | 13-0.5 | 470 | Wang et al., 2001 |
| G21 | 60 | Diaojiaohaizi Lake | 41.30 | 112.35 | 1800 | Lake | Yes | 30 | 309 | [14]C | 4U | 11.5-2.5 | 95 | Song et al., 1996 |

| | | | | | | | | | | | | | |
|---|---|---|---|---|---|---|---|---|---|---|---|---|---|
| G22 | 112 | Hulun Nur Lake_1995 | 49.28 | 117.40 | 544 | Lake | No | 233900 | 27286 | $^{14}$C | 7U | 19-0.5 | 190 | Yang et al., 1995 |
| G22 | 113 | Hulun Nur Lake_2006 | 49.13 | 117.51 | 545 | Lake | Yes | 233900 | 27286 | $^{14}$C | 13E | 11-0 | 65 | Wen et al., 2010 |
| G23 | 314 | Derput | 57.03 | 124.12 | 700 | Bog | Yes | 1 | 56 | $^{14}$C | 1A+4C | 11.7-0.8 | 210 | Andreev and Klimanov, 1991 |
| G23 | 409 | Suollakh | 57.05 | 123.85 | 811 | Bog | Yes | - | - | $^{14}$C | 8C | 12.8-3.7 | 180 | Andreev et al., 1991 |
| G24 | 379 | Nuochaga Lake | 61.30 | 129.55 | 260 | Lake | Yes | 120 | 618 | $^{14}$C | 4E | 6.5-0 | 140 | Andreev and Klimanov, 1989 |
| G24 | 307 | Chabada Lake | 61.98 | 129.37 | 290 | Lake | Yes | 210 | 818 | $^{14}$C | 15U | 13-0 | 110 | Andreev and Klimanov, 1989 |
| G25 | 305 | Boguda Lake | 63.67 | 123.25 | 120 | Lake | Yes | 2500 | 2821 | $^{14}$C | 7E | 10.9-0.4 | 180 | Andreev et al., 1989 |
| G25 | 344 | Khomustakh Lake | 63.82 | 121.62 | 120 | Lake | Yes | 440 | 1183 | $^{14}$C | 9E | 12.3-0.1 | 170 | Andreev et al., 1989 |
| G25 | 368 | Madjaga Lake | 64.83 | 120.97 | 160 | Lake | Yes | 1440 | 2141 | LSC | 7E | 8.2-0.2 | 120 | Andreev and Klimanov, 1989 |
| G25 | 302 | Billyakh Lake | 65.30 | 126.78 | 340 | Lake | Yes | 1678 | 2311 | $^{14}$C | 7A | 14.1-0 | 180 | Müller et al., 2009 |
| G25 | 426 | Lake Billyakh_PG1755 | 65.27 | 126.75 | 340 | Lake | Yes | 1634 | 2281 | $^{14}$C | 1A+10E | 50.6-0.2 | 470 | Müller et al., 2010 |
| G26 | 440 | Lake Kyutyunda_PG2022 | 69.63 | 123.65 | 66 | Lake | Yes | 468 | 1220 | $^{14}$C | 10E | 10.8-0.3 | 360 | Biskaborn et al., 2016 |
| G27 | 435 | Khocho | 71.05 | 136.23 | 6 | Bog | Yes | 10 | 178 | $^{14}$C | 1C | 10.4-0.4 | 300 | Velichko et al., 1994 |
| G27 | 441 | Samandon | 70.77 | 136.25 | 10 | Bog | Yes | 100 | 564 | $^{14}$C | 3A+8C+4E | 7.9-0.2 | 280 | Velichko et al., 1994 |
| G28 | 299 | Barbarina Tumsa | 73.57 | 123.35 | 10 | Bog | Yes | - | - | $^{14}$C | 4C | 4.9-0.3 | 240 | Andreev et al., 2004 |
| G28 | 373 | Lake Nikolay | 73.67 | 124.25 | 35 | Lake | Yes | 1500 | 2185 | $^{14}$C | 6A | 12.5-0 | 600 | Andreev et al., 2004 |
| G28 | 318 | Dolgoe Ozero | 71.87 | 127.07 | 12 | Lake | Yes | 84 | 517 | $^{14}$C | 1A+9B | 15.3-0 | 210 | Pisaric et al., 2001 |
| G29 | 152 | Maili | 42.87 | 122.88 | 155 | Bog | No | - | - | $^{14}$C | 5A | 3-0 | 115 | Ren and Zhang, 1997 |
| G29 | 54 | Dashan | 44.88 | 124.85 | 200 | Bog | Yes | - | - | $^{14}$C | 5U | 7.5-1 | 160 | Xia et al., 1993 |
| G29 | 240 | Xiaonan | 43.88 | 125.22 | 209 | Bog | Yes | - | - | $^{14}$C | 5U | 5.5-0 | 290 | Wang and Xia, 1988 |

| | | | | | | | | | | | | | |
|---|---|---|---|---|---|---|---|---|---|---|---|---|---|
| G29 | 197 | Shuangyang | 43.45 | 125.75 | 215 | Bog | Yes | - | - | $^{14}$C | 12E | 2.5-0 | 30 | Qiu et al., 1981 |
| G29 | 34 | Charisu | 42.95 | 122.35 | 249 | Bog | Yes | - | - | $^{14}$C | 10A | 5.5-0 | 170 | Li Y.H. et al., 2003b |
| G29 | 463 | Jingbo Lake | 43.91 | 128.75 | 350 | Lake | Yes | 9500 | 5499 | $^{14}$C+LSC | 3E+4 | 8.8-0 | 40 | Li et al., 2011 |
| G29 | 96 | Harbaling | 43.63 | 129.20 | 600 | Bog | Yes | - | - | $^{14}$C | 3U | 3-0 | 150 | Xia, 1988b |
| G29 | 122 | Jinchuan | 42.35 | 126.38 | 620 | Bog | Yes | - | - | $^{14}$C | 7A | 5.5-0 | 105 | Li Y.H. et al., 2003a |
| G29 | 72 | Erhailongwan Lake | 42.30 | 126.37 | 724 | Lake | Yes | 30 | 309 | $^{14}$C | 2A+14E | 22-0 | 760 | Liu Y.Y. et al., 2008 |
| G29 | 288 | Sihailongwan Lake | 42.28 | 126.60 | 797 | Lake | Yes | 41 | 360 | $^{14}$C+varve | 40A | 16.9-0.2 | 47 | Stebich et al., 2015 |
| G29 | 94 | Hani | 42.21 | 126.52 | 899 | Bog | Yes | 1800 | 2394 | $^{14}$C | 1C | 9.5-0 | 455 | Qiao, 1993 |
| G29 | 37 | Chichi Lake | 42.03 | 128.13 | 1800 | Bog | Yes | 0 | 40 | $^{14}$C | 1C | 1-0 | 140 | Xu et al., 1994 |
| G30 | 21 | Belaya Skala | 43.25 | 134.57 | 4 | Bog | Yes | - | - | $^{14}$C | 2A+1C | 6.5-3 | 250 | Korotky et al., 1980 |
| G30 | 36 | Chernyii Yar | 43.18 | 134.43 | 4 | Bog | Yes | - | - | $^{14}$C | 4C | 10-0.5 | 260 | Korotky et al., 1980 |
| G30 | 218 | Tikhangou | 42.83 | 132.78 | 4 | Bog | Yes | - | - | $^{14}$C | 5U | 12-0 | 500 | Korotky et al., 1980 |
| G30 | 5 | Amba River | 43.32 | 131.82 | 5 | Bog | Yes | - | - | $^{14}$C | 1A+1C+1U | 5-2.5 | 300 | Korotky et al., 1980 |
| G30 | 186 | Ryazanovka | 42.83 | 131.37 | 6 | Bog | Yes | - | - | $^{14}$C | 7A | 6-0.5 | 540 | Shilo, 1987 |
| G30 | 171 | Ovrazhnyii | 43.25 | 134.57 | 8 | Bog | Yes | - | - | $^{14}$C | 3A | 7-1 | 200 | Shilo, 1987 |
| G30 | 175 | Peschanka | 43.30 | 132.12 | 12 | Bog | Yes | - | - | $^{14}$C | 3U | 22-11 | 965 | Anderson et al., 2002 |
| G30 | 464 | Xingkai Lake | 45.21 | 132.51 | 69 | Lake | Yes | 419000 | 36520 | $^{14}$C+Pb/Cs | 3E | 28.5-0 | 150 | Ji et al., 2015 |
| G31 | 220 | Tongjiang | 47.65 | 132.50 | 49 | Bog | Yes | - | - | $^{14}$C | 5C | 6-0 | 130 | Zhang and Yang, 2002 |
| G31 | 40 | Chuangye | 48.33 | 134.47 | 50 | Bog | Yes | - | - | $^{14}$C | 3U | 12-1 | 400 | Xia, 1988a |
| G31 | 161 | Minzhuqiao | 47.53 | 133.87 | 52 | Bog | Yes | - | - | $^{14}$C | 4U | 6.5-0.5 | 420 | Xia, 1988a |
| G31 | 180 | Qindeli | 47.88 | 133.67 | 52 | Bog | Yes | - | - | $^{14}$C | 1F+7U | 13.5-0.5 | 380 | Xia, 1988a |
| G31 | 18 | Beidawan | 48.13 | 134.70 | 60 | Bog | Yes | 8 | 157 | $^{14}$C | 3U | 5.5-0.5 | 350 | Xia, 1988a |
| G31 | 234 | Wuchanghai | 47.22 | 127.33 | 200 | Bog | Yes | - | - | $^{14}$C | 9E | 7-0 | 250 | Xia, 1988b |
| G31 | 212 | Tangbei | 48.35 | 129.67 | 486 | Bog | Yes | - | - | $^{14}$C | 2A | 5.5-1 | 160 | Xia, 1996 |

| G32 | 418 | Venyukovka-3 | 47.12 | 138.58 | 5 | Bog | Yes | - | - | $^{14}$C | 1A+2C | 5.8-3.2 | 140 | Korotky et al., 1980 |
|-----|-----|--------------|-------|--------|---|-----|-----|---|---|------|-------|------|-----|------|
| G32 | 417 | Venyukovka-2 | 47.03 | 138.58 | 6 | Bog | Yes | - | - | $^{14}$C | 1A+1C | 3.6-0.4 | 140 | Korotky et al., 1980 |
| G32 | 384 | Oumi | 48.22 | 138.40 | 990 | Bog | Yes | - | - | $^{14}$C | 5C | 2.6-0.4 | 80 | Anderson et al., 2002 |
| G32 | 382 | Opasnaya River | 48.23 | 138.48 | 1320 | Bog | Yes | - | - | $^{14}$C | 7C | 13.3-6.7 | 360 | Korotky et al., 1988 |
| G33 | 335 | Il'inka Terrace | 47.97 | 142.17 | 3 | Bog | Yes | - | - | $^{14}$C | 2C+1F | 2.6-1.1 | 360 | Korotky et al., 1997 |
| G33 | 371 | Mereya River | 46.62 | 142.92 | 4 | Bog | Yes | - | - | $^{14}$C | 2C+2F | 42.0-0.8 | 1530 | Anderson et al., 2002 |
| G33 | 401 | Sergeevskii | 49.23 | 142.08 | 6 | Bog | Yes | - | - | $^{14}$C | 8A+1C | 8.4-2.2 | 110 | Korotky et al., 1997 |
| G34 | 332 | Gurskii Peat | 50.07 | 137.08 | 15 | Bog | Yes | - | - | $^{14}$C | 7C | 13.1-1.5 | 380 | Korotky, 1982 |
| G34 | 453 | Gur Bog | 50.00 | 137.05 | 35 | Bog | No | - | - | $^{14}$C | 13C | 22.1-0 | 340 | Mokhova et al., 2009 |
| G34 | 223 | Tuqiang | 52.23 | 122.80 | 400 | Bog | Yes | - | - | $^{14}$C | 10A+14E+8F | 3-1 | 125 | Xia, 1996 |
| G34 | 398 | Selitkan-2 | 53.22 | 135.03 | 1300 | Bog | Yes | - | - | $^{14}$C | 4C | 6.4-1.9 | 260 | Volkov and Arkhipov, 1978 |
| G34 | 397 | Selitkan-1 | 53.22 | 135.05 | 1320 | Bog | Yes | - | - | $^{14}$C | 6C | 7.9-0 | 140 | Korotky et al., 1985 |
| G35 | 443 | Two-Yurts Lake_PG1856-3 | 56.82 | 160.04 | 275 | Lake | Yes | 1168 | 1928 | $^{14}$C | 5A | 6.0-2.8 | 140 | Hoff et al., 2015 |
| G35 | 444 | Two-Yurts Lake_PG1857-2 | 56.82 | 160.07 | 275 | Lake | Yes | 1168 | 1928 | $^{14}$C | 5A | 2.5-0.1 | 130 | Hoff et al., 2015 |
| G35 | 445 | Two-Yurts Lake_PG1857-5 | 56.82 | 160.07 | 275 | Lake | Yes | 1168 | 1928 | $^{14}$C | 5A | 4.4-2.5 | 120 | Hoff et al., 2015 |
| G35 | 455 | Lake Sokoch | 53.25 | 157.75 | 495 | Lake | Yes | 41 | 363 | $^{14}$C | 8E | 9.7-0.3 | 250 | Dirksen et al., 2012. |
| G36 | 330 | Glukhoye Lake | 59.75 | 149.92 | 10 | Bog | Yes | - | - | $^{14}$C | 5C | 9.4-3.4 | 1000 | Lozhkin et al., 1990 |
| G36 | 312 | Chistoye Lake | 59.55 | 151.83 | 91 | Bog | Yes | - | - | $^{14}$C | 5C | 7.0-0 | 540 | Anderson ey al., 1997 |
| G36 | 363 | Lesnoye Lake | 59.58 | 151.87 | 95 | Lake | Yes | 13 | 200 | $^{14}$C | 8A | 15.5-0 | 400 | Anderson et al., 1997 |
| G36 | 388 | Pepel'noye Lake | 59.85 | 150.62 | 115 | Lake | Yes | 0 | 18 | $^{14}$C | 2A | 4.3-0 | 180 | Lozhkin et al., 2000 |
| G36 | 290 | Alut Lake | 60.14 | 152.31 | 480 | Lake | Yes | 63 | 448 | $^{14}$C | 16A+9B | 50.4-0 | 430 | Anderson et al., 1998 |

| G36 | 391 | Podkova Lake | 59.96 | 152.10 | 660 | Lake | Yes | 114 | 602 | $^{14}$C | 5A | 6.0-0 | 220 | Anderson et al., 1997 |
|---|---|---|---|---|---|---|---|---|---|---|---|---|---|---|
| G36 | 370 | Maltan River | 60.88 | 151.62 | 735 | Bog | Yes | - | - | $^{14}$C | 4A+7C | 12.0-9.4 | 120 | Lozhkin and Glushkova, 1997 |
| G36 | 411 | Taloye Lake | 61.02 | 152.33 | 750 | Lake | Yes | 16 | 227 | $^{14}$C | 7A | 10.3-0 | 290 | Lozhkin et al., 2000 |
| G36 | 323 | Elikchan 4 Lake | 60.75 | 151.88 | 810 | Lake | Yes | 329 | 1023 | $^{14}$C | 16U | 55.5-0 | 440 | Lozhkin and Anderson, 1995 |
| G36 | 331 | Goluboye Lake | 61.12 | 152.27 | 810 | Lake | Yes | 12 | 192 | $^{14}$C | 11A+2B | 9.7-0 | 240 | Lozhkin et al., 2000 |
| G36 | 470 | Julietta Lake | 61.34 | 154.56 | 880 | Lake | Yes | 11 | 189 | $^{14}$C | 2A+4E+1I | 36.1-1.4 | 270 | Anderson et al., 2010 |
| G36 | 321 | Elgennya Lake | 62.08 | 149.00 | 1040 | Lake | Yes | 455 | 1204 | $^{14}$C | 6A | 16.0-0 | 310 | Lozhkin et al., 1996 |
| G37 | 405 | Smorodinovoye Lake | 64.77 | 141.12 | 800 | Lake | Yes | 27 | 293 | $^{14}$C | 6A+5F | 27.1-0 | 360 | Anderson et al., 1998 |
| G37 | 416 | Vechernii River | 63.28 | 147.75 | 800 | Bog | Yes | - | - | $^{14}$C | 1F | 14.4-0.1 | 380 | Anderson et al., 2002 |
| G37 | 338 | Jack London Lake | 62.17 | 149.50 | 820 | Lake | Yes | 1213 | 1965 | $^{14}$C | 7F | 19.5-0.2 | 320 | Lozhkin et al., 1993 |
| G37 | 406 | Sosednee Lake | 62.17 | 149.50 | 822 | Lake | Yes | 82 | 510 | $^{14}$C | 4E+1F | 26.3-0 | 640 | Lozhkin et al., 1993 |
| G37 | 393 | Rock Island Lake | 62.03 | 149.59 | 849 | Lake | Yes | 5 | 124 | $^{14}$C | 2E | 6.6-0 | 470 | Lozhkin et al., 1993 |
| G37 | 381 | Oldcamp Lake | 62.04 | 149.59 | 853 | Lake | Yes | 7 | 150 | $^{14}$C | 2E | 3.7-0 | 370 | Anderson, unpublished |
| G37 | 329 | Gek Lake | 63.52 | 147.93 | 969 | Lake | Yes | 2392 | 2759 | $^{14}$C | 8A+1B | 9.6-0 | 440 | Stetsenko, 1998 |
| G37 | 433 | Figurnoye Lake | 62.10 | 149.00 | 1053 | Lake | Yes | 439 | 1182 | $^{14}$C | 4A | 1.3-0 | 30 | Lozhkin et al., 1996 |
| G38 | 353 | Kuropatoch'ya_Kurop7 | 70.67 | 156.75 | 7 | Bog | Yes | - | - | $^{14}$C | 3C | 5.7-0.4 | 760 | Anderson et al., 2002 |
| G38 | 354 | Kuropatoch'ya_Kurpeat | 69.97 | 156.38 | 47 | Bog | Yes | - | - | $^{14}$C | 1A+4C | 11.7-7.5 | 430 | Lozhkin and Vazhenina, 1987 |
| G39 | 322 | Elgygytgyn Lake | 67.50 | 172.10 | 496 | Lake | No | 9503 | 5500 | polarity | - | 20.2-1.5 | 650 | Melles et al., 2012 |
| G39 | 325 | Enmynveem_mammoth | 68.17 | 165.93 | 400 | Bog | Yes | 50 | 399 | $^{14}$C | 2C+2F | 36.4-9.3 | 2470 | Lozhkin et al., 1988 |
| G39 | 326 | Enmyvaam River | 67.42 | 172.08 | 490 | Bog | Yes | 18 | 239 | $^{14}$C | 1A+4C | 10.6-4.3 | 630 | Lozhkin and Vazhenina, 1987 |
| G39 | 324 | Enmynveem River | 68.25 | 166.00 | 500 | Bog | Yes | - | - | $^{14}$C | 4C | 10.7-4.0 | 420 | Anderson et al., 2002 |

| | | | | | | | | | | | | | |
|---|---|---|---|---|---|---|---|---|---|---|---|---|---|
| G40 | 454 | Malyi Krechet Lake | 64.80 | 175.53 | 32 | Lake | Yes | 125 | 630 | [14]C | 12A | 9.6-0 | 400 | Lozhkin and Anderson, 2013 |
| G40 | 456 | Melkoye Lake | 64.86 | 175.23 | 36 | Lake | Yes | 1870 | 2440 | [14]C | 21E | 39.1-0 | 1260 | Lozhkin and Anderson, 2013 |
| G40 | 460 | Sunset Lake | 64.84 | 175.30 | 36 | Lake | Yes | 240 | 874 | [14]C | 7A | 14.0-0 | 260 | Lozhkin and Anderson, 2013 |
| G40 | 333 | Gytgykai Lake | 63.42 | 176.57 | 102 | Lake | Yes | 99 | 561 | [14]C | 1A+8E | 32.3-0 | 470 | Lozhkin et al., 1998 |
| G40 | 457 | Patricia Lake | 63.33 | 176.50 | 121 | Lake | Yes | 40 | 357 | [14]C | 3A+7E | 19.1-0 | 290 | Anderson and Lozhkin, 2015 |
| G41 | 436 | Konergino | 65.90 | -178.90 | 10 | Bog | Yes | - | - | [14]C | 1C | 9.8-0 | 900 | Ivanov et al., 1984 |
| G41 | 365 | Lorino | 65.50 | -171.70 | 12 | Bog | Yes | - | - | [14]C | 3C | 17.9-5.1 | 850 | Ivanov, 1986 |
| G41 | 317 | Dlinnoye Lake | 67.75 | -178.83 | 280 | Lake | Yes | 71 | 476 | [14]C | 3A | 1.3-0 | 130 | Anderson et al., 2002 |
| G41 | 431 | Dikikh Olyenyeii Lake | 67.75 | -178.83 | 300 | Lake | Yes | 64 | 450 | [14]C | 1A+4C | 50.3-0 | 1050 | Anderson et al., 2002 |
| G42 | 427 | Blossom Cape | 70.68 | 178.95 | 6 | Bog | Yes | - | - | [14]C | 1C | 13.8-0.2 | 3400 | Oganesyan et al., 1993 |
| G42 | 420 | Wrangle Island_Jack London Lake | 70.83 | -179.75 | 7 | Lake | Yes | 69 | 469 | [14]C | 5A+1E | 16.1-0.3 | 790 | Lozhkin et al., 2001 |
| G42 | 419 | Wrangel Island | 71.17 | -179.75 | 200 | Bog | Yes | - | - | [14]C | 17A+3C | 13.7-10.2 | 110 | Lozhkin et al., 2001 |

LSC: liquid-scintillation counting; A: terrestrial plant macrofossil; B: non-terrestrial plant macrofossil; C: peat; D: pollen; U: unknown; E: total organic matter from silt; F: animal remains or shell; G: charcoal; H: $CaCO_3$; I: tephra.

**Appendix 3** Slight percentage changes for five major plant taxa reconstructed by REVEALS model with different bog radii (5 m, 10 m, 20 m, 50 m, 100 m, 200 m
and 500 m).

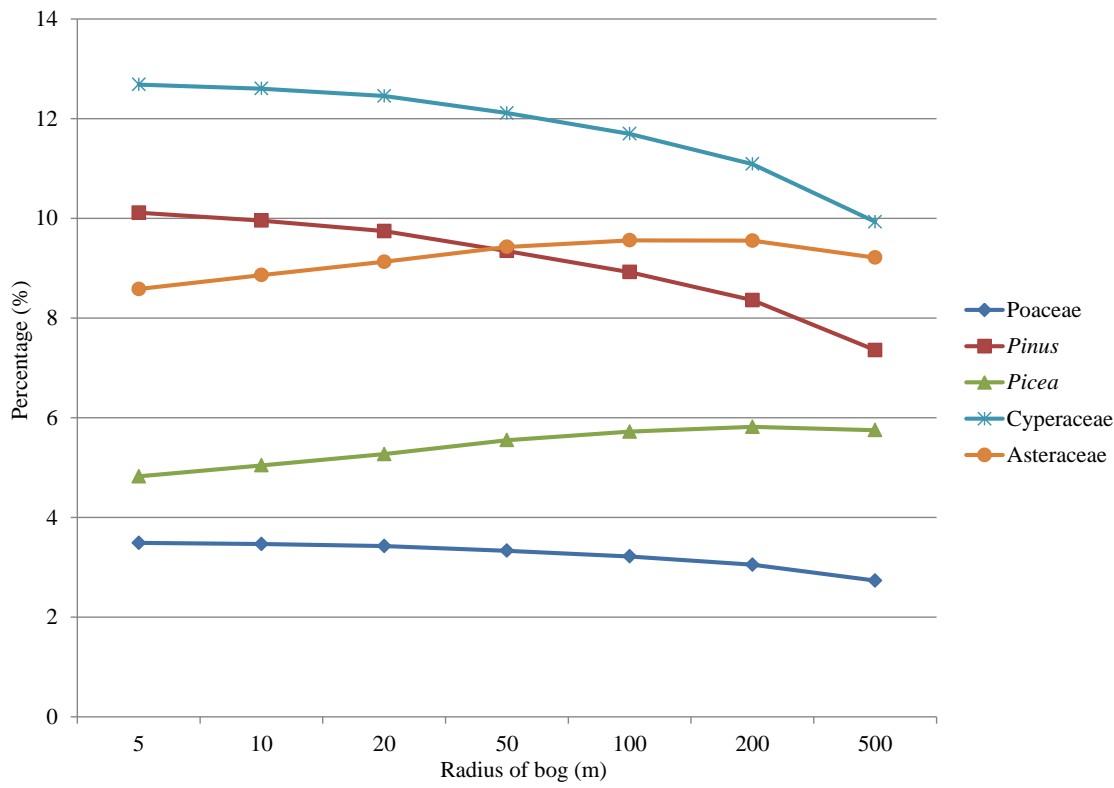




**Appendix 4** Pollen Productivity Estimates (PPEs) with their standard errors (SEs) for 27 pollen taxa from 20 study areas. Estimates where SE $\geq$ PPE were
excluded from the calculation of mean PPE and are shown in italics.

| Country | Poland | Russia | Sweden | Sweden | Swiss | Swiss | Switzerland | Sweden | Finland | Estonia |
|---|---|---|---|---|---|---|---|---|---|---|
| Region | Białowieża Forest | Khatanga region | Southern Sweden | Southern Sweden | Swiss Plateau | Alps | Jura Mountains | west-central | Fennoscandia | Southeast |
| sample type | Moss | Moss | Moss | Moss | Lake | Trap | Moss | Moss | Moss | Lake |
| Reference | Baker et al., 2016 | Niemeyer et al., 2015 | Broström et al., 2004 | Sugita et al., 1999 | Soepboer et al., 2007 | Sjögren et al., 2008 | Mazier et al., 2008 | von Stedingk et al., 2008 | Räsänen et al., 2007 | Poska et al., 2011 |
| Model | ERV-3 | ERV-2 | ERV-3 | ERV-3 | ERV-3 | - | ERV-1 | ERV-3 | ERV-3 | ERV-1 |
| Poaceae | 1 (0.00) | 1 (0.00) | 1 (0.00) | 1 (0.00) | 1 (0.00) | 1 (0.00) | 1 (0.00) | 1 (0.00) | 1 (0.00) | 1 (0.00) |
| *Abies* | | | | | 9.92 (2.86) | | 3.83 (0.37) | | | |
| *Pinus* | 23.12 (0.24) | | | 5.66 (0.00) | 1.35 (0.45) | 9 (0.00) | | 21.58 (2.87) | 8.4 (1.34) | 5.07 (0.06) |
| *Picea* | | | | 1.76 (0.00) | 0.57 (0.16) | 0.5 (0.00) | 7.1 (0.2) | 2.78 (0.21) | | 4.73 (0.13) |
| *Larix* | | *0.00009 (0.1)* | | | | 1.4 (0.00) | | | | |
| *Alnus*_tree | 15.95 (0.66) | | | 4.2 (0.14) | | 20 (0.00) | | | | 13.93 (0.15) |
| *Betula*_tree | 13.94 (0.23) | | | 8.87 (0.13) | 2.42 (0.39) | | | 2.24 (0.2) | 4.6 (0.7) | 1.81 (0.02) |
| *Juglans* | | | | | | | | | | |
| *Fraxinus* | | | | 0.67 (0.03) | 1.39 (0.21) | | | | | |
| *Quercus* | 18.47 (0.10) | | | 7.53 (0.08) | 2.56 (0.39) | | | | | 7.39 (0.2) |
| *Tilia* | 0.98 (0.03) | | | 0.8 (0.03) | | | | | | |
| *Ulmus* | | | | | | | | | | |
| *Alnus*_shrub | | 6.42 (0.42) | | | | | | | | |
| *Betula*_shrub | | 1.8 (0.26) | | | | | | | | |
| *Carpinus* | 4.48 (0.03) | | | | 4.56 (0.85) | | | | | |
| *Corylus* | 1.35 (0.05) | | | 1.4 (0.04) | 2.58 (0.25) | | | | | |

| | Czech<br>Central Bohemia | Norway<br>South | Greenland<br>Southern | England<br>Calthorpe | England<br>Wheatfen | Germany<br>Brandenburg | China<br>Tibetan Plateau | China<br>Xilinhaote | China<br>Shandong | China<br>Changbai Mt. |
|---|---|---|---|---|---|---|---|---|---|---|
| *Salix* | | 0.03 (0.03) | | 1.27 (0.31) | | | 0.09 (0.03) | | | 2.31 (0.08) |
| Ericaceae | | 0.33 (0.03) | | | | | 0.07 (0.04) | | | |
| *Ephedra* | | | | | | | | | | |
| Cyperaceae | | 0.53 (0.06) | 1 (0.16) | | | 0.68 (0.01) | 0.89 (0.03) | 0.002 (0.0022) | | 1.23 (0.09) |
| *Artemisia* | | | | | | | | | | 3.48 (0.19) |
| Chenopodiaceae | | | | | | | | | | |
| Asteraceae | | | 0.24 (0.06) | | 0.17 (0.03) | | | | | |
| *Thalictrum* | | | | | | | | | | |
| Ranunculaceae | | | 3.85 (0.72) | | | | | | | |
| Caryophyllaceae | | | | | | | | | | |
| Brassicaceae | | | | | | | | | | |


| Country | Czech | Norway | Greenland | England | England | Germany | China | China | China | China |
|---|---|---|---|---|---|---|---|---|---|---|
| Region | Central Bohemia | South | Southern | Calthorpe | Wheatfen | Brandenburg | Tibetan Plateau | Xilinhaote | Shandong | Changbai Mt. |
| sample type | Moss | Lake | Moss | Moss | Moss | Lake | Lake | Soil | moss | Moss |
| Reference | Abraham and Kozáková, 2012 | Hjelle and Sugita, 2012 | Bunting et al., 2013 | Bunting et al., 2005 | Bunting et al., 2005 | Matthias et al., 2012 | Wang and Herzschuh, 2011 | Xu et al., 2014 | Li et al., 2017 | Li et al., 2015 |
| Model | ERV-1 | ERV-3 | ERV-1 | Average | Average | allFIDage_ERV3 | ERV-2 | ERV2 | ERV-3 | - |
| Poaceae | 1 (0.00) | 1 (0.00) | 1 (0.00) | 1 (0.00) | 1 (0.00) | 1 (0.00) | 1 (0.00) | 1 (0.00) | 1 (0.00) | |
| *Abies* | | | | | | | | | | |
| *Pinus* | 6.17 (0.41) | 5.73 (0.07) | | | | 5.2 (0.00) | | | 8.96 (0.23) | 15.2079 (0.489) |
| *Picea* | | 1.2 (0.04) | | | | 1.456 (0.05) | | | | |
| *Larix* | | | | | | 8.06 (0.32) | | | | 1.47 (0.19) |

| | | | | | | | | | | |
|---|---|---|---|---|---|---|---|---|---|---|
| *Alnus*_tree | 2.56 (0.32) | 3.22 (0.22) | | 10.564 (0.00) | 4.028 (0.00) | 14.248 (0.22) | | | | |
| *Betula*_tree | | | 3.7 (0.4) | 9.804 (0.00) | | 8.84 (0.34) | | | | 24.65 (0.73) |
| *Juglans* | | | | | | | | 0.3 (0.05) | 9.49 (0.44) | |
| *Fraxinus* | 1.11 (0.09) | | | 1.14 (0.00) | 0.076 (0.00) | 6.188 (0.12) | | | | 3.72 (0.68) |
| *Quercus* | 1.76 (0.2) | 1.3 (0.1) | | 7.6 (0.00) | 7.6 (0.00) | 1.976 (0.03) | | | 4.89 (0.16) | |
| *Tilia* | 1.36 (0.26) | | | | | 1.352 (0.04) | | | | 0.78 (0.19) |
| *Ulmus* | | | | | | | 11.5 (1.09) | | 1 (0.31) | 6.85 (1.71) |
| *Alnus*_shrub | | | | | | | | | | |
| *Betula*_shrub | | | 1.4 (0.05) | | | | | | | |
| *Carpinus* | | | | | | 8.684 (0.09) | | | | |
| *Corylus* | | | | | 1.216 (0.00) | | | | | |
| *Salix* | 1.19 (0.12) | 0.62 (0.11) | 0.8 (0.002) | 1.748 (0.00) | 2.736 (0.00) | | | | | |
| Ericaceae | | | | | | | | | | |
| *Ephedra* | | | | | | | 0.96 (0.14) | | | |
| Cyperaceae | | 1.37 (0.21) | 0.95 (0.05) | | | | 0.65 (0.4) | 0.94 (0.079) | 0.21 (0.07) | |
| *Artemisia* | 2.77 (0.39) | | | | | | 3.2 (0.6) | 11.21 (0.31) | 24.7 (0.36) | |
| Chenopodiaceae | 4.28 (0.27) | | | | | | 5.3 (1.1) | 6.74 (0.79) | | |
| Asteraceae | | | | | | | | 0.39 (0.16) | 1.06 (0.21) | |
| *Thalictrum* | | | 4.65 (0.3) | | | | | 3.06 (0.42) | | |
| Ranunculaceae | | | 1.95 (0.1) | | | | | | | |
| Caryophyllaceae | | | 0.6 (0.05) | | | | | | | |
| Brassicaceae | | | | | | | | 7.48 (0.33) | 0.89 (0.18) | |




**Appendix 5** Number of pollen records from large lakes (≥390 m radius; represented by L), small lakes (<390 m radius; represented by S), and bogs (B) for each
site-group used to run REVEALS for each time slice. For example, site-group G6 has 2 large lake records, 1 small lake record, and 2 bog records at 4 ka
(represented by 2L1S2B).

| Grou | 0 ka | 0.2 ka | 0.5 ka | 1 ka | 2 ka | 3 ka | 4 ka | 5 ka | 6 ka | 7 ka | 8 ka | 9 ka | 10 ka | 11 ka | 12 ka | 14 ka | 21 ka | 25 ka | 40 ka |
|---|---|---|---|---|---|---|---|---|---|---|---|---|---|---|---|---|---|---|---|
| G1 | 1L | 1L | 1L | - | 1L | - | 1L | 1L | 1L | 1L | 1L | 1L | - | - | - | - | - | - | - |
| G2 | 6B | 1S6B | 1S6B | 1S6B | 1S4B | 2S6B | 2S6B | 1S4B | 2S2B | 1S | 2S | 2S | 1S2B | 1S2B | 1S2B | 2B | - | - | - |
| G3 | 4B | 4B | 8B | 8B | 6B | 8B | 8B | 8B | 8B | 6B | 6B | 4B | 4B | 4B | - | - | - | - | - |
| G4 | - | 1L | - | 1L | 1L | 1L | 1L | 1L | 1L | 1L2B | 1L2B | 1L | 1L | 1L | 1L | - | - | - | - |
| G5 | 4S4B | 4S4B | 4S4B | 4S4B | 1S4B | 1S4B | 1S4B | 1S2B | 1S2B | 1S2B | - | - | - | - | - | - | - | - | - |
| G6 | 2L1S2B | 1L1S2B | 2L1S4B | 2L1S4B | 2L1S4B | 1L1S2B | 2L1S2B | 1S | 1L1S | 1L2B | 1L2B | 1L2B | 2B | 2B | 2B | 2B | - | - | - |
| G7 | 4B | 10B | 12B | 12B | 1L12B | 1L12B | 1L10B | 1L10B | 1L10B | 1L10B | 6B | 8B | 8B | 1L6B | 2B | - | 2B | - | 2B |
| G8 | 2B | 2B | 4B | 4B | 2B | 4B | 6B | 8B | 8B | 8B | 6B | 4B | 4B | 4B | 2B | - | - | - | - |
| G9 | 4B | 4B | 6B | 6B | 4B | 6B | 6B | 2B | 6B | 4B | 8B | 8B | 8B | 8B | 4B | 2B | - | - | - |
| G10 | 1L | 1L | 1L | 1L | 2B | 1L2B | 1L4B | 1L6B | 1L8B | 1L6B | 1L6B | 1L6B | 1L4B | 1L2B | 1L | 1L | 1L | - | - |
| G11 | 2L1S | 2L1S | 2L1S | 2L1S | 1L1S | 2L1S | 1L1S | 1L1S | 2L1S | 2L1S | 2L | 2L | 1L | 1L | 2L | 1L | 1L | 1L | - |
| G12 | 6L1S2B | 5L1S2B | 5L1S2B | 6L1S2B | 5L1S2B | 3L1S2B | 5L1S2B | 4L1S2B | 4L2B | 4L2B | 5L2B | 4L | 4L | 3L | 4L | 1L | - | - | - |
| G13 | 1L | 1L | 1L | 2L | 2L | 2L | 2L | 2L | 2L | 2L | 2L | 2L | 1L | 1L | 1L | 1L | - | - | - |
| G14 | 4L | 1L | 4L | 4L1S | 5L1S | 5L2S | 5L1S | 4L1S | 3L1S | 4L2S | 4L2S | 4L2S | 3L1S | 4L2S | 4L1S | 3L2S | - | - | - |
| G15 | 1L | 2L | 2L | 2L | 2L | 3L | 3L | 3L | 3L | 2L | 2L | 3L | 1L | 3L | 3L | 2L | 1L | - | - |
| G16 | 1L | - | 2L | - | 2L | 2L | 2L | 1L | 1L | 2L | 2L | 2L | 2L | 2L | 3L | 1L | 2L | 3L | - |
| G17 | - | - | - | - | 1L | 1L | - | 1L | 1L | 1L | 1L | 1L | - | 1L | - | - | - | - | - |
| G18 | 2L2S | 3L1S | 2L2S | 4L2S | 2L1S | 4L1S | 5L1S | 4L1S | 4L1S | 4L | 5L | 4L1S | 2L1S | 3L1S | 4L | 2L | - | - | - |
| G19 | - | 1L | - | 1L | 1L | - | 1L | 1L | - | - | - | - | - | - | - | - | - | - | - |
| G20 | 6L6B | 4L4B | 6L8B | 5L1S6B | 6L1S8B | 5L8B | 5L6B | 5L1S6B | 5L1S6B | 5L1S4B | 4L4B | 4L2B | 5L2B | 5L2B | 6L2B | 5L2B | 2L2B | 2L2B | 1L |
| G21 | 4L1S2B | 2L1S2B | 4L1S2B | 4L1S2B | 3L1S2B | 4L2S4B | 4L2S4B | 3L2S4B | 3L1S4B | 4L1S2B | 5L1S4B | 4L1S2B | 5L1S2B | 6L1S | 5L1S | 1L | - | - | - |
| G22 | 1L | 1L | 2L | 2L | 2L | 2L | 2L | 2L | 2L | 2L | 2L | 2L | 2L | 2L | 1L | 1L | - | - | - |
| G23 | - | - | - | 2B | 2B | 2B | 4B | 4B | 4B | 4B | 4B | 4B | 4B | 4B | 4B | - | - | - | - |
| G24 | 2L | 2L | 2L | 2L | 2L | 2L | 2L | 2L | 2L | 1L | 1L | 1L | 1L | 1L | 1L | - | - | - | - |
| G25 | 1L | 4L | 4L | 4L | 5L | 5L | 5L | 5L | 5L | 4L | 4L | 3L | 3L | 4L | 2L | 2L | 1L | 1L | 1L |
| G26 | 1L | - | 1L | 1L | 1L | 1L | 1L | 1L | 1L | - | - | 1L | 1L | 1L | - | - | - | - | - |

| | | | | | | | | | | | | | | | | | | | |
|---|---|---|---|---|---|---|---|---|---|---|---|---|---|---|---|---|---|---|---|
| G27 | - | 2B | 4B | 4B | 4B | 2B | 4B | 4B | 4B | 4B | 4B | 2B | - | - | - | - | - | - | - |
| G28 | 2L | 2B | 2L2B | 1L2B | 2L2B | 1L2B | 2L2B | 2B | 2L | 1L | 2L | 2L | 2L | 2L | 1L | 1L | - | - | - |
| G29 | 1L1S10B | 1L1S14B | 1L2S14B | 1L1S16B | 1L1S16B | 1L2S16B | 1L1S10B | 1L2S10B | 1L1S4B | 1L2S4B | 1L2S2B | 1L1S2B | 2S | 2S | 1S | 1S | 1S | 1S | 1S |
| G30 | 1L | 1L2B | 1L6B | 1L4B | 1L8B | 1L8B | 1L6B | 1L10B | 1L8B | 1L8B | 1L4B | 1L4B | 1L2B | 1L4B | 1L4B | 1L2B | 1L4B | 1L4B | 4B |
| G31 | 2B | 2B | 10B | 14B | 12B | 14B | 10B | 12B | 10B | 4B | 2B | 4B | 2B | 4B | 4B | - | - | - | - |
| G32 | - | - | 4B | 4B | 4B | 2B | 2B | 2B | 2B | 2B | 2B | 2B | 2B | 2B | - | - | - | - | - |
| G33 | - | - | - | 4B | 2B | 2B | 4B | 2B | 4B | 2B | 2B | - | - | 2B | 2B | 2B | 2B | 2B | 2B |
| G34 | 4B | 4B | 4B | 6B | 10B | 8B | 8B | 6B | 6B | 6B | 6B | 4B | 4B | 4B | 4B | 2B | 2B | - | - |
| G35 | - | 1L1S | 1L1S | 1L1S | 1L1S | 2L1S | 1L | 1L1S | 1L | 1S | 1S | 1S | 1S | - | - | - | - | - | - |
| G36 | 4L4S2B | 2L2S | 4L3S | 4L4S | 4L4S | 4L5S | 4L4S | 3L2S2B | 4L2S | 2L4S4B | 3L4S2B | 3L4S | 2L4S2B | 3L2S2B | 2L2S2B | 2L2S | 2L1S | 2L1S | 2L1S |
| G37 | 3L3S | 2L1S2B | 3L1S2B | 1L3S2B | 1L3S2B | 2L3S2B | 1L3S2B | 2L2S2B | 3L2S2B | 3L1S | 1L1S | 2L | 2L1S | 2L1S | 1L1S | 2L1S | 2L1S | 1L1S | - |
| G38 | - | - | 2B | 2B | - | 2B | 2B | 2B | 2B | - | 2B | 2B | 2B | 2B | 2B | - | - | - | - |
| G39 | - | - | - | - | - | 1L | 1L2B | - | 1L4B | 2B | 1L4B | 1L4B | 2B | 1L4B | 1L | 1L | 1L2B | 2B | 2B |
| G40 | 4L1S | 1L | 2L1S | 3L1S | 3L1S | 2L | 1S | 2L | 2L1S | 1S | 1L1S | 3L1S | 2L | 2L | 3L | 2L | 2L1S | 1L | 1L |
| G41 | 2L2B | 1L | 1L | 1L | 2B | 2B | - | 4B | - | 4B | 4B | 4B | 2B | 1L2B | 2B | 1L2B | 1L | 1L | 1L |
| G42 | - | 1L2B | - | 1L | 1L | 1L | 1L | - | - | - | - | - | 1L | 1L2B | 1L4B | 1L4B | - | - | - |








**Appendix 6** Cluster diagram of the site-groups based on the plant functional type
dataset

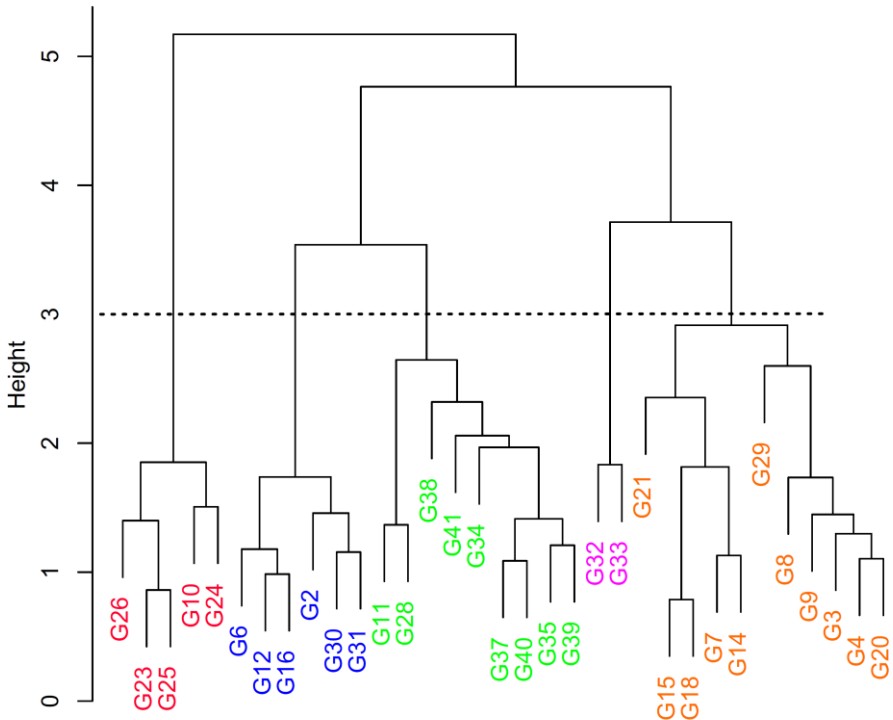
























**Appendix 7** Map of the study area showing the geographic locations mentioned in the text.

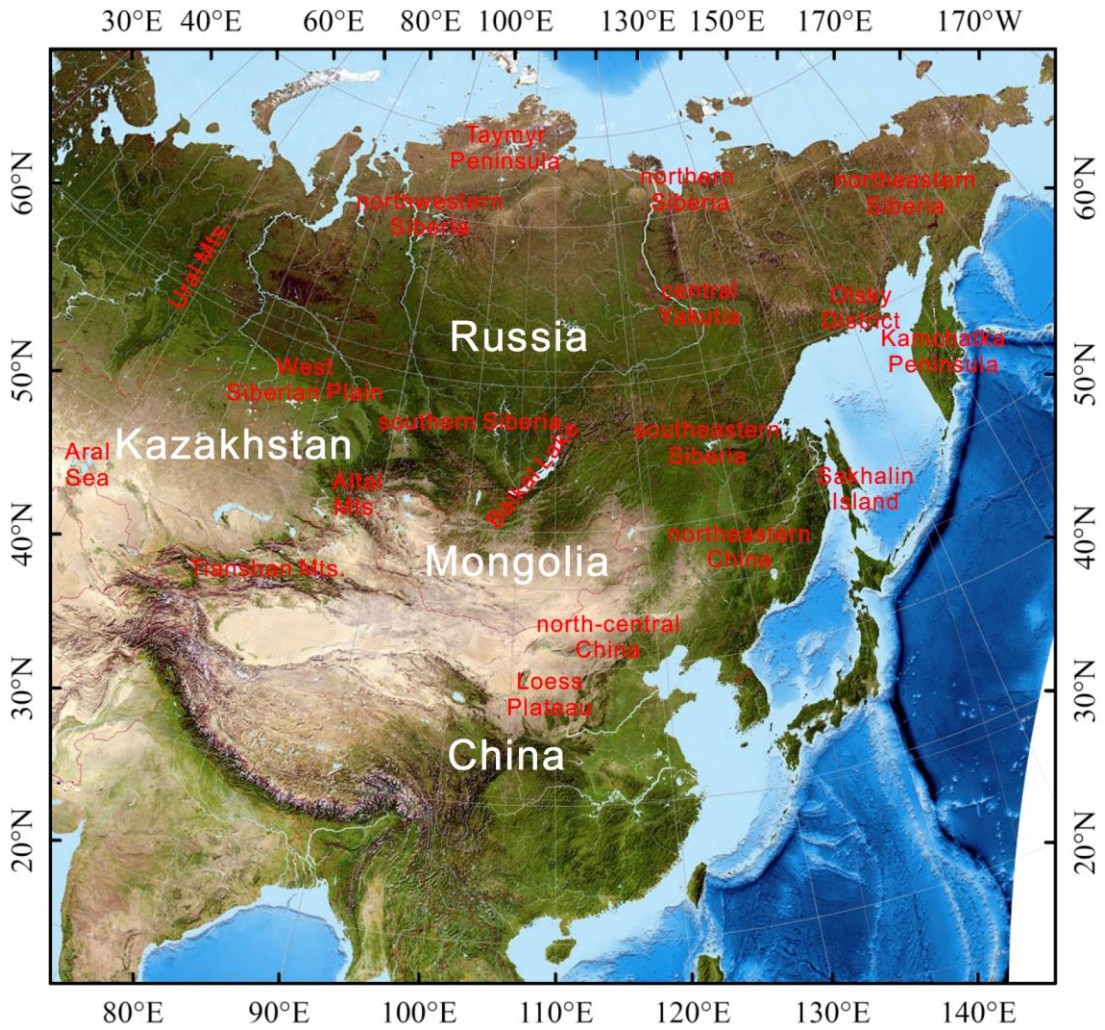


**Appendix 8** Selected examples of standard errors for seven plant functional type (PFT)
reconstructions at site-groups G21, G20, and G36 at 6 ka.

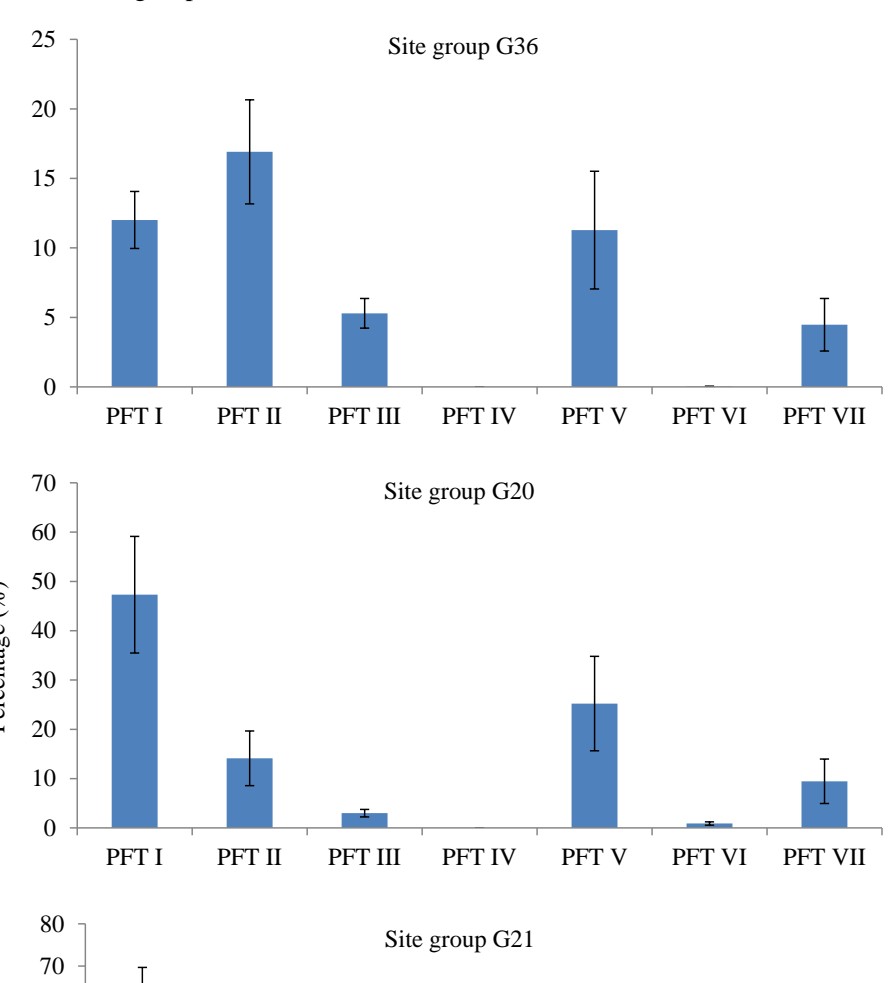

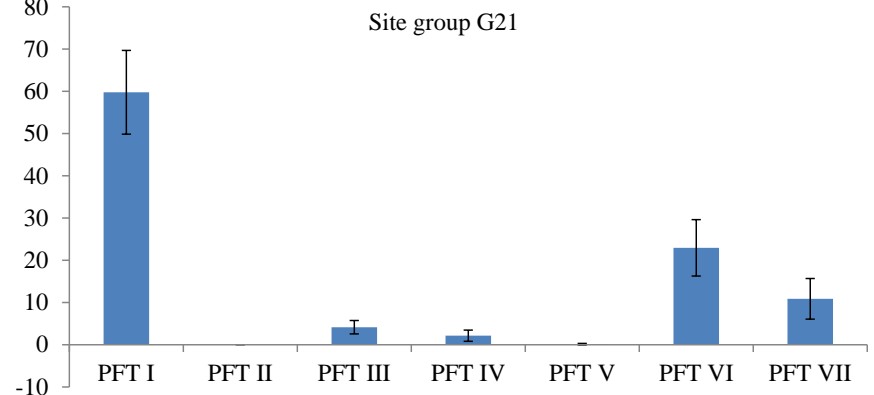










**Appendix 9** Proxy-based climate reconstructions from the Northern Hemisphere and insolation
variations during the last 40 cal ka BP discussed in the paper. NGRIP: the North Greenland
Ice-Core Project (Andersen et al. 2004); Sanbao cave (Cheng et al. 2016); Alkenone-derived
sea-surface temperatures (SST) from deep-sea cores SU8118 and MD952042 (Pailler and Bard
2002); solar insolation in July at 60 °N (Laskar et al. 2004).

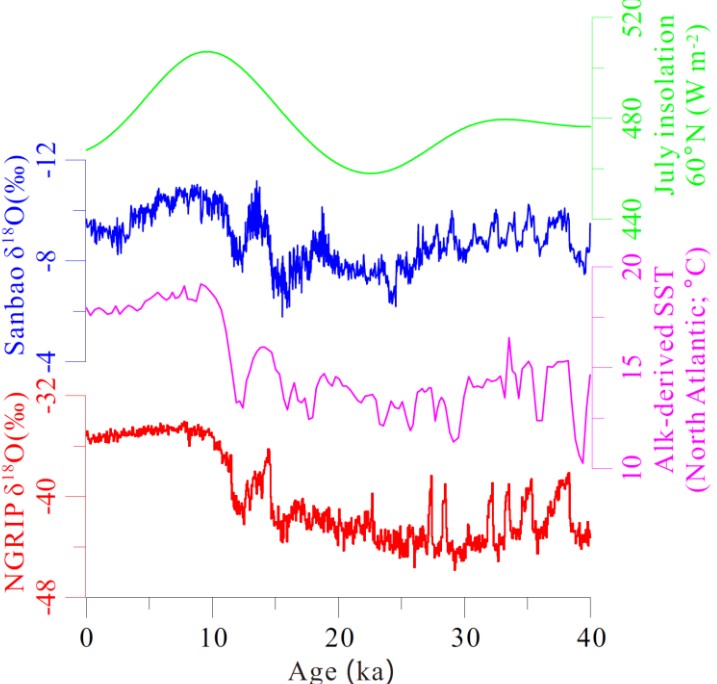
