# Peer review of "Pollen-based quantitative land-cover reconstruction for northern Asia covering"

_Climate of the Past, 2018_

## Referee Comment (RC1) · Anonymous Referee #1 · 26 Nov 2018

This paper focuses on past land-cover changes based on pollen data. This study considers the REVEALS model to quantify plant abundances (27 taxa) and related PFTs at the spatial scale of northern Asia and a temporal scale covering the last 40 ka. It is a great paper providing substantial information for the scientific community. One of the major interests of this work is the application of the REVEALS model at such spatial scale as it has been done over the last years for Europe. This work is therefore a good contribution for environmental and climate sciences by providing additional quantitative land-cover reconstructions at the continental scale of the Northern Hemisphere, and this is particularly critical for climate modelling. I would highly recommend this manuscript for publication in Climate of The Past. I do have comments and suggestions hereafter.

Major comments

1- I strongly recommend to revise the presentation of the results (core text) and figures 2 and 3. The authors use 42 groups for the REVEALS reconstructions and this makes difficult to observe major trends in vegetation changes, at least as it is presented here. To improve this, I have the following suggestions. First, I would only situate the pollen archives in figure 1; this figure has already too much information. Second, I suggest to group pollen archives into meaningful data such as vegetation zones, however to be objective I recommend to use the results from the cluster analysis, i.e 5 cluster groups. The 42 groups can be shown in an appendix when in the core text only the PFT results for the 5 cluster groups would presented. This would mean only one figure 2 rather than the duplication of figure 2. This would make easy the reading of the result sections and more useful/meaningful the figures. I further recommend to show the results for the turnover (entire time period and 5 cluster groups); this can be a new figure 3. This means that the result section should be partially revised, and the results of the cluster analysis should be shown in a specific graph; there are no graphs about the results of the cluster analysis in the present version of the manuscript.

2- Through the manuscript, the authors wrote that vegetation changes in northern Asia within the Holocene are "minor with slight changes in PFTs" (e.g. lines 212-213). This conclusion is based on the fact that the turnover is high during the early-Holocene and the numerical analysis based on constrained hierarchical clustering and the broken-stick model provide a timing of the primary change mostly during the early-Holocene. However, changes in PFT abundances can be high (e.g. G7 shows around 20 % fewer abundance of PFT VII between 2 ka and 1 ka). I would suggest to clearly define what the turnover and the timing of the primary change really mean. I am wondering if the identification of a primary change necessary implies that no other changes of similar importance can occurred more recently.

3- The conclusion is too short and do not show the potential of the present study.

Minor comments

1- Why the turnover has been calculated from PFTs (see lines 203-206) rather than pollen types? This might explain differences between the turnover results in Europe and this study. Lower turnover in the present study than the ones in Europe might be related to the use of less variables here (PFTs) than in Europe (pollen types). The discussion lines 411-417 need to be revised by considering this issue.

2- The selection of RPP values is critical for this study. The relevance in the present study of using RPPs that have been obtained in Europe or other environmental and ecological conditions in Asia needs to be discussed in more details. Furthermore, more than the 20 RPP studies that the authors refer to have been published, and if they are taken into account they would increase a lot the uncertainties related to the choice of the specific RPPs that have been used by the authors, e.g. Chenopodiaceae, Artemisia and Compositae RPPs in Li et al. (Frontiers in Plant Science 2018). Different species within the Chenopodiaceae family might result in different RPPs, although we do not know how much this play a role in RPP calculation. All of these issues need to be taken into account in the discussion section, specifically for lines 428-452.

3- The choice of PFT VII (steppe and forb tundra) might be misleading. Artemisia pollen type is included in PFT VI (arid-tolerant shrub and herb), however Artemisia is an important component of steppe vegetation. Furthermore, tundra vegetation is located north of the study region (vegetation zone A) when steppe are located more south (vegetation zone D), this considers the vegetation zones that the authors provide. It would probably be more relevant to relate steppe to Artemisia and therefore PFT VI. This would not affect the results. 4- Lines 148-151. Why the authors have selected the specific value of 100 m for all bogs?

5- Lines 193-195. This linear interpolation when it corresponds to a large time gap of missing time windows might be a source of uncertainties, and it would be good to further discussed this.

[Figure]

6- Lines 372-376. I suggest to be more specific about how the DNA information supports the results.

7- Lines 377-380. This sentence is not clear.

8- The authors should be consistent and use the term RPP through the manuscript; the term PPE can be found in several paragraphs.

9- I would suggest to add a global map in a corner of figure 1 to show the location of the study area. It is too much information and the reader can be "spatially lost".

10- There is here no discussion about land-use changes for the last millennia. I would be interested in how land-use can be discussed based on these PFTs; I expect that the land-use should have some influences on forest covers at some points (e.g. late-Holocene primary vegetation changes).

11- Can the authors give the REVEALS standard errors in an appendix to get an idea about how reliable the reconstructions are?

12- I suggest to move Table 3 to Appendix.

13- Lines 464-466. I suggest to give more information about what "riverine" really means here (erosions, water-runoff, temporal lake etc. . .) and how these processes might affect the results.

14- I would add to the discussion a short sentence about the assumed constant RPP values over the last 40ka.

15- Line 473. I would not use the term "observed" but rather "past". This to avoid a potential confusion with modern vegetation that can really be "observed".

16- Line 189. I think the "The end of moisture increase" is confusing or there is a mistake here.

17- The authors might add some climate information via a new summary figure. This

could be informative and useful to follow the discussion section.

18- Lines 504-505. I disagree with this conclusion. Vegetation–climate relationship can be "linear" and no strong effect observed at short time scales (e.g. few decades). It might just be a matter of time scales, i.e. long-term responses of vegetation.

---

## Referee Comment (RC2) · Anonymous Referee #2 · 3 Dec 2018

The aim of the study is to use pollen data for quantifying in very general terms the vegetation change in northern Asia over the last 40 ka. A key element in the study is the use the REVEALS method, which accounts for the differences in pollen productivity and dispersal, hence providing more robust abundance estimates than pollen percentage values. In general, I am in favour of suggesting that the paper can be accepted because it is based on a substantial dataset and presents results over a large area which has not been intensively investigated so far.

At the same time, I would urge the authors to amend the paper by making it clearer, more structured and more informative for readers. I found that large sections of the text, especially in Results and Discussion, were hard to follow. One reason is that there are too few references to figures and tables in the paper, making it hard to find

out whether the interpretations presented in the text were sound and really supported by the results. For example, on page 10, the first long paragraph presents many types of results, but the only reference to a figure is at the end of the paragraph "(e.g. G"9, G39; Fig. 2)". Similarly, the next paragraph begins "The turnover in PFT composition is <0.7 SD units in almost all site-groups, except G8 (0.88 SD), G9 (0.73 SD), and G24 (0.76 SD) indicating only slight vegetation change during the Holocene" – are these results shown somewhere in the paper? I did not find them in the figures or tables.

Most of the results in the paper are shown in Fig. 2., which is a very big figure, divided into three parts. It is not an easy figure to follow together with the text. My suggestion would be to section the figure to smaller parts, either as three separate figures (Figs. 2, 3, 4) or sub-panels (Fig. 2a, 2b, 2c). Fig. 1 shows the study regions and the datapoints, but there are too much data squeezed into the figure, so it is a bit messy.

While I understand the motivation of using the REVEALS method in the paper, I also notice that the spreads in the estimated PPE values for different pollen types are remarkable. This can be best seen by looking at the Appendix 2. Consequently, there must be an enormous error associated with these estimates, and that uncertainty should be kept in mind throughout the discussion and conclusions. In addition, in the paper, the PPE value used for Larix is 3.642. But in the Appendix 2 it is indicated that there are two earlier PPE estimates for Larix, 0.00009 and 1.4. What is the value 3.642 based on? Note also that both "RPP" and "PPE" are used as abbreviations for the term "pollen productivity estimates", for example in Table 2 and Appendix 2.

In addition to the use of the REVEALS method, the pollen types are converted to plant functional types (pft) for defining the vegetation types for the study period. After this conversion, the selected 27 pollen types were reduced to only seven pft. This is sometimes useful because it allows a very generalized presentation of past vegetation types, but it also influences the results of the vegetation turnover rate calculations, which the authors have carried out by applying DCCA with their pft data. This results is an extremely simplified measure, where the resulting turnover values includes errors that

stem from the uncertainty in defining the PPE values and from the heavy generalization involved in converting the pollen types to pfts. I would not therefore place too much emphasis for the resulting turnover calculations presented in Fig 3.

Table 2 shows that Corylus is assigned to the plant functional type group "boreal deciduous trees". A "temperate deciduous tree" would probably be more correct. And how did the authors handle the pollen types which belong to two different plant functional type groups (for example, Betula is in boreal deciduous trees and boreal shrubs)

Finally, the sites with pollen data were divided into 42 site groups. Each site groups includes many subregions, which are scattered around northern Asia. It remains unclear why such a subdivision was considered useful and how the site groups were defined. The description on page 8 says that "...we divided the 203 into 42 site groups, based on criteria on geographic location, vegetation type, climate and permafrost. This is a confusing description because one site groups can contain subregions from different parts of Asia, so it is hard to understand how they could have been defined on the basis of geographic location or climate, for example.
* * *

---

## Editor Comment (EC1) · Jonkers (Editor) · 13 Dec 2018

Thank you for your contribution to the inter-journal special issue 'Paleoclimate data synthesis and analysis of associated uncertainty' and for your help to promote open source paleoclimate science.

One of the goals of the special issue on 'Paleoclimate data synthesis and analysis of associated uncertainty' is to promote good data stewardship in paleoclimatology. Therefore, the data handling of all contributions to the special issue will be reviewed independently from the normal peer reviews and short comments. While we realise that this may lead to additional work on the authors' side, we believe that good data stewardship is essential to guarantee transparency and reproducibility of the results as

well as to promote the reuse of the data. The editors will be requesting evidence that all data presented in the submissions are made available freely and adhere to the FAIR concept (https://www.nature.com/articles/sdata201618). This applies to original data, as well as to data compilations and derived data products. Where relevant, authors are asked to adhere to the practice of attributing data to original authors through data citation and encouraged to share code used to treat original data and generate derived data products.

Specific comments for Pollen-based quantitative land-cover reconstruction for northern Asia during the last 40 ka by Cao et al.

In their data availability statement, the authors mention that the full data set will be made publicly available in a contribution to the journal Earth System Science Data. We are happy that the authors are willing to share their data. However, as far as we are aware, no such manuscript has been submitted to ESSD. This means that the data effectively remain unavailable.

Required additions:

As an alternative option to the submission of a new manuscript, we suggest that the authors make their data publicly available at this stage. If needed the data can be placed under embargo until acceptance of a revised manuscript, but reviewers and editor should have the possibility to access the data and evaluate the data handling. Also note that publication of a revised manuscript requires that the embargo be lifted and the data are publicly available. In this way the review process can continue uninterrupted and needs not to depend on publication of a second manuscript.

When making the data available, please ensure that full metadata are archived to facilitate straightforward use of the data. This would for instance include notes on the taxonomic harmonisation. In addition, please include raw pollen counts where available and include data citations where appropriate (e.g. for previously published data and the data being made available in this manuscript).

[Figure]

Further recommended additions:

Consider making the code used to analyse the data and to make the figures accessible on a public repository (GitHub, Zenodo).

If you have any questions about the data handling guidelines, please don't hesitate to contact any of the guest editors of the special issue 'Paleoclimate data synthesis and analysis of associated uncertainty'.

With kind regards,

Lukas Jonkers On behalf of the guest editorial team

---

## Author Comment (AC1) · 31 Mar 2019

Major comments

1, I strongly recommend to revise the presentation of the results (core text) and figures 2 and 3. The authors use 42 groups for the REVEALS reconstructions and this makes difficult to observe major trends in vegetation changes, at least as it is presented here. To improve this, I have the following suggestions. First, I would only situate the pollen archives in figure 1; this figure has already too much information.

*Our response:* **We agree with the comment about Figure 1. In the new version, we have separated it into two figures: one presents the vegetation and permafrost background of the study region, together with the locations of the site-groups as the new Figure 1; and the second presents the locations of the pollen records (with IDs of pollen sites) and has been moved into the appendices as Appendix 1. The old "Appendix 1" (information for all pollen records) is now Appendix 2.**

Second, I suggest to group pollen archives into meaningful data such as vegetation zones, however

to be objective I recommend to use the results from the cluster analysis, i.e 5 cluster groups. The 42 groups can be shown in an appendix when in the core text only the PFT results for the 5 cluster groups would presented. This would mean only one figure 2 rather than the duplication of figure 2. This would make easy the reading of the result sections and more useful/meaningful the figures. I further recommend to show the results for the turnover (entire time period and 5 cluster groups); this can be a new figure 3. This means that the result section should be partially revised, and the results of the cluster analysis should be shown in a specific graph; there are no graphs about the results of the cluster analysis in the present version of the manuscript.

*Our response:* **We revised the presentation of the results. We now present the changes of the site-groups that belong to one cluster as a separate figure. However, the results could not be further summarized. The REVEALS approach is suitable to reconstruct the *regional* land-cover change assuming that these regions have similar temporal change patterns. Previous reconstructions in Europe were at a $1° \times 1°$ spatial scale (c. 100 km $\times$ 100 km; Trondman et al., 2016). In Northern Asia, there are not enough available pollen data, thus we had to consider the available data at a larger spatial scale. We feel that summarizing the results into only 5 groups would be an over-simplification.**

**The PFT dataset for the 34 site-groups covering the period between 12 and 1 ka is a three-way dataset. We calculated the distance set for the 34 site-groups using the tsclust function and then performed a simple hclust analysis. The hclust analysis cannot produce cluster means for each cluster; hence we cannot present a general pattern for the five clusters. In the new version, we present the cluster diagram as Appendix 5.**

**Regarding the turnover calculation (DCCA), we cannot perform the DCCA for the entire time span because most site-groups do not cover the entire time span and DCCA cannot deal with missing values. Hence, we can only perform the cluster and turnover analyses for the**

**Holocene.**

**In the new version, we present the reconstructed results for each cluster one by one, replacing the old Figure 2 (which presented reconstructed results by site-group). The new Figure 2 includes 5 sub-figures, separated by modern vegetation: A) warm temperate forest margin zone, B) cool-temperate mixed forest, C) dark taiga forest, D) light taiga forest, and E) the tundra-taiga ecotone, which is consistent with the "Results" part.**

2, Through the manuscript, the authors wrote that vegetation changes in northern Asia within the Holocene are "minor with slight changes in PFTs" (e.g. lines 212-213). This conclusion is based on the fact that the turnover is high during the early-Holocene and the numerical analysis based on constrained hierarchical clustering and the brokenstick model provide a timing of the primary change mostly during the early-Holocene. However, changes in PFT abundances can be high (e.g. G7 shows around 20 % fewer abundance of PFT VII between 2 ka and 1 ka). I would suggest to clearly define what the turnover and the timing of the primary change really mean. I am wondering if the identification of a primary change necessary implies that no other changes of similar importance can occurred more recently.

*Our response:* **We conclude "minor with slight changes in PFTs" during the period between 12 and 1 cal ka BP because the turnover is *low* throughout the period *and* because some of the primary change does not pass the broken-stick test during the constrained hierarchical clustering. Most of the insignificant or significant primary changes occur in the early Holocene, hence we conclude that the most important vegetation changes occur in the early Holocene. In the new version, we define what turnover and the timing of the primary change mean.**

Lines 202-208

*"Constrained hierarchical clustering (using chclust function in rioja package version 0.9-15.1;*

*Juggins, 2018) was used to determine the timing of primary vegetation changes (i.e. the first split) in each site-group. A change was considered to be significant when the split passed the broken-stick test. The amount of PFT compositional change (turnover) through time during the period between 12 and 1 ka was estimated by detrended canonical correspondence analysis (DCCA) for each site-group (ter Braak, 1986) using CANOCO 4.5 (ter Braak and Šmilauer, 2002)."*

3, The conclusion is too short and do not show the potential of the present study.

***Our response:* we have revised the conclusion in the new version.**

Lines 558-570

*"Regional vegetation based on pollen data has been estimated using the REVEALS model for northern Asia during the last 40 ka. Relatively closed land cover was replaced by open landscapes in northern Asia during the transition from MIS 3 to the last glacial maximum. Abundances of woody components increase again from the last deglaciation or early Holocene. Pollen-based REVEALS estimates of plant abundances should be a more reliable reflection of the vegetation as pollen may overestimate turnover and indicates that the vegetation was quite stable during the Holocene as only slight changes in the abundances of PFTs were recorded rather than mass expansion of new PFTs. From comparisons of our results with other data we infer that climate change is likely the primary driving factor for vegetation changes on a glacial-interglacial scale. However, the extension of evergreen conifer trees since ca. 8–7 ka throughout Siberia could reflect vegetation-climate disequilibrium at a long-term scale caused by the interaction of climate, vegetation, fire, and permafrost."*

Minor comments

1, Why the turnover has been calculated from PFTs (see lines 203-206) rather than pollen types? This might explain differences between the turnover results in Europe and this study. Lower turnover in the present study than the ones in Europe might be related to the use of less variables here (PFTs) than in Europe (pollen types). The discussion lines 411-417 need to be revised by

considering this issue.

***Our response*: We used the PFT dataset to calculate turnover because both the REVEALS model and our manuscript focus on the changes in PFTs rather than taxa. We have revised the phrase "in the abundance of major taxa rather than by invasions of new taxa" and replaced it by "in the abundance of PFTs rather than by invasions of new PFTs", In addition, we have added a discussion about why there is relatively low turnover in North Asia compared to Europe.**

Lines 24-28

*"Reconstructed regional plant-functional type (PFT) components for each site-group are generally consistent with modern vegetation in that vegetation changes within the regions are characterized by minor changes in the abundance of PFTs rather than by invasions of new PFTs, particularly during the Holocene."*

Lines 428-432

*"The fewer parameters used in the turnover calculations for northern Asia (PFTs) compared to Europe (pollen taxa) is a potential reason for the lower turnover obtained in this study. In addition, the PPE-based transformation from pollen percentages to plant abundances may reduce the strength of vegetation changes (Wang and Herzschuh, 2011)."*

2, The selection of PPE values is critical for this study. The relevance in the present study of using PPEs that have been obtained in Europe or other environmental and ecological conditions in Asia needs to be discussed in more details. Furthermore, more than the 20 PPE studies that the authors refer to have been published, and if they are taken into account they would increase a lot the uncertainties related to the choice of the specific PPEs that have been used by the authors, e.g. Chenopodiaceae, *Artemisia* and Compositae PPEs in Li et al. (Frontiers in Plant Science 2018). Different species within the Chenopodiaceae family might result in different PPEs, although we

do not know how much this play a role in PPE calculation. All of these issues need to be taken into account in the discussion section, specifically for lines 428-452.

*Our response*: **We agree with the reviewer. The quality of the PPEs is most relevant for the reliability of reconstruction. However, hitherto only 20 PPE records from Eurasia have been published. We applied a consistent approach to calculate the PPEs from the published values. i.e. we used all PPE records of different species within one family or genus in the calculation of the mean PPE for the family or genus. Furthermore, we argue "the regional differences in the PPE for each taxon are small compared to the large between-taxa differences ". We have added some further discussion about the reliability of PPEs in the new version.**

Lines 154-155

*"We included these PPE values for various species in the mean PPE calculation for their family or genus."*

Lines 451-456

*"The available PPEs were estimated from various environmental and ecological settings, which might cause regional differences in each PPE. Also, PPEs of different species within one family or genus were included in our mean-PPE calculation for the family or genus, ignoring the inter-species differences. Both these aspects can cause uncertainty in the mean PPE to some extent."*

3- The choice of PFT VII (steppe and forb tundra) might be misleading. *Artemisia* pollen type is included in PFT VI (arid-tolerant shrub and herb), however *Artemisia* is an important component of steppe vegetation. Furthermore, tundra vegetation is located north of the study region (vegetation zone A) when steppe are located more south (vegetation zone D), this considers the vegetation zones that the authors provide. It would probably be more relevant to relate steppe to

*Artemisia* and therefore PFT VI. This would not affect the results.

***Our response***: ***Artemisia* pollen at high abundance is found in arid central Asia, while quite low abundances are found in northern Asia (Siberia). As mentioned by the reviewer, *Artemisia* is an important component in arid steppe community. However, in order to separate the tundra forb and the steppe forb, we had to separate the arid-tolerant forb and the wet-favouring forb. The name for PFT VII was misleading and so we have changed it to "grassland and tundra forb" in Table 2.**

4- Lines 148-151. Why the authors have selected the specific value of 100 m for all bogs?

***Our response***: **We had performed test-runs that showed that the different bog radii (i.e. 5 m, 10 m, 20 m, 50 m, 100 m, 200 m and 500 m) did not significantly affect the REVEALS estimates, hence a standard radius of 100 m was set for all bogs. We have added the explanation into the new version and a figure as Appendix 3.**

Lines 148-151

*"A test-run showed that using different bog radii (i.e. 5 m, 10 m, 20 m, 50 m, 100 m, 200 m and 500 m) did not significantly affect the REVEALS estimates (Appendix 3), hence a standard (moderate size) radius of 100 m was set for all bogs."*

5- Lines 193-195. This linear interpolation when it corresponds to a large time gap of missing time windows might be a source of uncertainties, and it would be good to further discussed this.

***Our response***: **We agree with the reviewer and have added some discussion.**

Lines 485-488

*"In addition, the linear interpolation of pollen abundances for time windows with few pollen data*

*might be another source of uncertainty, particularly for the late Pleistocene and its broad time windows (Table 1)."*

6- Lines 372-376. I suggest to be more specific about how the DNA information supports the results.

*Our response***: Agree and added.**

Lines 386-394

*"During the late Pleistocene (40, 25, 21, 14 ka), steppe PFT abundance was high in central Yakutia and north-eastern Siberia (e.g. G25, G36, G37, G39, G40, G41), which may reflect the expansion of tundra-steppe, consistent with results from ancient sediment DNA which reveal abundant forb species during the period between 46 and 12.5 ka on the Taymyr Peninsula (Jørgensen et al., 2012). The tundra-steppe was replaced by light taiga in southern Siberia and by tundra in northern Siberia at the beginning of Holocene or the last deglaciation, which is consistent with ancient DNA results (forbs-dominated steppe-tundra; Willerslev et al., 2014)."*

7- Lines 377-380. This sentence is not clear.

*Our response***: Agree and improved.**

Lines 395-397

*"During the Holocene, reconstructed land cover for each site-group is generally consistent with their modern vegetation. The slight vegetation changes are represented by changes in PFT abundances rather than by changes in PFT presence/absence."*

8- The authors should be consistent and use the term PPE through the manuscript; the term PPE can be found in several paragraphs.

*Our response***: Agree and modified.**

9- I would suggest to add a global map in a corner of figure 1 to show the location of the study area. It is too much information and the reader can be "spatially lost".

*Our response*: **Agree and added.**

10- There is here no discussion about land-use changes for the last millennia. I would be interested in how land-use can be discussed based on these PFTs; I expect that the land-use should have some influences on forest covers at some points (e.g. late-Holocene primary vegetation changes).

*Our response*: **Yes, land cover could be modified by humans during the late Holocene. However, in our study area, human impact on vegetation is not very clear, because the regional land cover was reconstructed from a multi-record combination. Hence, we cannot discuss the land-use history.**

11- Can the authors give the REVEALS standard errors in an appendix to get an idea about how reliable the reconstructions are?

*Our response*: **We will upload the reconstruction datasets including standard errors to Pangaea after this manuscript is accepted. In the new version, we select three site-groups to illustrate the reasonable standard errors of reconstruction as Appendix 8.**

Lines 439-441

*"We consider the REVEALS-based regional vegetation-cover estimations in this study as generally reliable with reasonable standard errors (Appendix 8) thanks to the thorough selection of records with high quality pollen data and reliable chronologies."*

12- I suggest to move Table 3 to Appendix.

*Our response*: **Agree and done.**

13- Lines 464-466. I suggest to give more information about what "riverine" really means here (erosions, water-runoff, temporal lake etc: : :) and how these processes might affect the results.

*Our response*: **Agree. We have replaced "riverine" by "water-runoff".**

Lines 488-491

*"Finally, pollen signals from certain sites and during certain periods may be of water-runoff origin rather than aerial origin violating the assumption of the REVEALS-model that pollen is transported by wind."*

14- I would add to the discussion a short sentence about the assumed constant PPE values over the last 40ka.

*Our response*: **Agree and done.**

Lines 138-140

*"The REVEALS model assumes the PPEs of pollen taxa are constant variables over the target period, and requires parameter inputs including sediment basin radius (m), fall speed of pollen grain (FS, m/s), and PPE with standard error (SE; Sugita, 2007)."*

15- Line 473. I would not use the term "observed" but rather "past". This to avoid a potential confusion with modern vegetation that can really be "observed".

*Our response*: **Agree. We had replaced "observed" by "pollen-based reconstructed".**

Lines 493-496

*"On a glacial-interglacial scale, pollen-based reconstructed land-cover changes in northern Asia are generally consistent with the global climate signal (e.g. sea-surface temperature: Pailler and Bard, 2002; ice-core: Andersen et al., 2004; solar insolation: Laskar et al., 2004; cave deposits: Cheng et al., 2016; Appendix 9)."*

16- Line 189. I think the "The end of moisture increase" is confusing or there is a mistake here.

*Our response*: **Agree. We have deleted "The end of".**

17- The authors might add some climate information via a new summary figure. This could be informative and useful to follow the discussion section.

*Our response*: **Agree. We have prepared a figure as Appendix 9.**

18- Lines 504-505. I disagree with this conclusion. Vegetation–climate relationship can be "linear" and no strong effect observed at short time scales (e.g. few decades). It might just be a matter of time scales, i.e. long-term responses of vegetation.

**Our response: Agree and modified.**

Lines 568-570

*"However, the extension of evergreen conifer trees since ca. 8–7 ka throughout Siberia could reflect vegetation-climate disequilibrium at a long-term scale caused by the interaction of climate, vegetation, fire, and permafrost."*

---

## Author Comment (AC2) · 31 Mar 2019

**Response to the interactive comment on "Pollen-based quantitative land-cover reconstruction for northern Asia covering the last 40 ka"**

**Anonymous Referee #2**

The aim of the study is to use pollen data for quantifying in very general terms the vegetation change in northern Asia over the last 40 ka. A key element in the study is the use the REVEALS method, which accounts for the differences in pollen productivity and dispersal, hence providing more robust abundance estimates than pollen percentage values. In general, I am in favor of suggesting that the paper can be accepted because it is based on a substantial dataset and presents results over a large area which has not been intensively investigated so far.

At the same time, I would urge the authors to amend the paper by making it clearer, more structured and more informative for readers. I found that large sections of the text, especially in Results and Discussion, were hard to follow. One reason is that there are too few references to figures and tables in the paper, making it hard to find out whether the interpretations presented in the text were sound and really supported by the results. For example, on page 10, the first long paragraph presents many types of results, but the only reference to a figure is at the end of the paragraph "(e.g. G"9,G39; Fig. 2)". Similarly, the next paragraph begins "The turnover in PFT composition is <0.7 SD units in almost all site-groups, except G8 (0.88 SD), G9 (0.73 SD), and G24 (0.76 SD) indicating only slight vegetation change during the Holocene" – are these results shown somewhere in the paper? I did not find them in the figures or tables.

Most of the results in the paper are shown in Fig. 2., which is a very big figure, divided into three parts. It is not an easy figure to follow together with the text. My suggestion would be to section the figure to smaller parts, either as three separate figures (Figs. 2, 3, 4) or sub-panels (Fig. 2a, 2b, 2c). Fig. 1 shows the study regions and the datapoints, but there are too much data squeezed into the figure, so it is a bit messy.

*Our response*: **Agree. We have re-organized the manuscript, particularly Figures 1 and 2. In the new version, we present the reconstructed results for each cluster one by one, replacing the old Figure 2 (which presented reconstructing results by site-group). The new Figure 2**

**includes 5 sub-figures (Figure 2A–2E) separated by modern vegetation: warm temperate forest margin zone, cool-temperate mixed forest, dark taiga forest, light taiga forest, and the tundra-taiga ecotone, which is consistent with the discussion part. We have added more references for Figure 2 in the main text. The new Figure 1 presents only the vegetation and permafrost background of site-groups. We put the map of pollen-data locations and IDs in the appendices as the new Appendix 1.**

While I understand the motivation of using the REVEALS method in the paper, I also notice that the spreads in the estimated PPE values for different pollen types are remarkable. This can be best seen by looking at the Appendix 2. Consequently, there must be an enormous error associated with these estimates, and that uncertainty should be kept in mind throughout the discussion and conclusions. In addition, in the paper, the PPE value used for Larix is 3.642. But in the Appendix 2 it is indicated that there are two earlier PPE estimates for Larix, 0.00009 and 1.4. What is the value 3.642 based on? Note also that both "RPP" and "PPE" are used as abbreviations for the term "pollen productivity estimates", for example in Table 2 and Appendix 2.

*Our response:* **We have standardised the abbreviation for "relative pollen productivity estimates" as "PPE", and replaced "RPP" by "PPE" in the text. The old Appendix 2 (Appendix 4 in the new version) is quite a large table to present the PPE records for the 27 pollen taxa, so we separated it into two parts. There is another PPE value for *Larix* in the second part of the table of the new Appendix 4.**

In addition to the use of the REVEALS method, the pollen types are converted to plant functional types (pft) for defining the vegetation types for the study period. After this conversion, the selected 27 pollen types were reduced to only seven pft. This is sometimes useful because it allows a very generalized presentation of past vegetation types, but it also influences the results of

the vegetation turnover rate calculations, which the authors have carried out by applying DCCA with their pft data. This results is an extremely simplified measure, where the resulting turnover values includes errors that stem from the uncertainty in defining the PPE values and from the heavy generalization involved in converting the pollen types to pfts. I would not therefore place too much emphasis for the resulting turnover calculations presented in Fig 3.

*Our response:* **The first reviewer also mentioned this issue. Merging from pollen taxa to PFT does ignore some vegetation signal but also reduces the noise in the regional vegetation patterns. In our manuscript, we focus on the general and regional signal of vegetation change. Nevertheless, a spatial comparison of turnover in our study is still necessary to show the spatial variation in the density of vegetation changes. In addition, the conclusion of "minor with slight changes in PFTs" during the period between 12 and 1 cal ka BP, is not only based on the low turnover, but also on the insignificant primary change for many site-groups, which are also relative to some extent to summarize the taxa in the PFTs.**

Table 2 shows that *Corylus* is assigned to the plant functional type group "boreal deciduous trees". A "temperate deciduous tree" would probably be more correct. And how did the authors handle the pollen types which belong to two different plant functional type groups (for example, *Betula* is in boreal deciduous trees and boreal shrubs).

*Our response*: **The assignment of pollen taxa to PFT was completed follwing previous biome reconstruction literature (Tarasov et al., 1998, 2000; Bigelow et al., 2003; Ni et al., 2010). Pollen taxon "*Corylus*" was assigned to "cool-temperate cold-deciduous malacophyll broad-leaved tree or shrub" by Tarasov et al. (1998; 2000) and Ni et al. (2010), and to two PFTs - "boreal cold-deciduous malacophyll broad-leaved tree" and "temperate (spring-frost tolerant) cold-deciduous malacophyll broad-leaved tree" - by Bigelow et al. (2003; biome**

**reconstruction for north of 55 ˚N). In this study, pollen taxon "*Corylus*" occurs only in 10**

**sites at more than 3% abundance and most of these sites are north of 50 ˚N. Hence, we**

**assigned it to "boreal deciduous trees".**

**As well as *Betula*, the genus *Alnus* also has this problem of including both tree and shrub**

**species within one genus. It is quite difficult to separate the pollen grains of genus *Betula* and**

***Alnus* into tree or shrub type by optical identification. In our dataset, many pollen records**

**did not separate the tree and shrub type for the two genera, although some did. The**

**undifferentiated *Betula* and *Alnus* grains were assigned to trees in our study, while pollen**

**taxa with clear statements about their identity were separated based on these statements. We**

**have modified Table 1.**

Finally, the sites with pollen data were divided into 42 site groups. Each site groups includes many subregions, which are scattered around northern Asia. It remains unclear why such a subdivision was considered useful and how the site groups were defined. The description on page 8 says that "we divided the 203 into 42 site groups, based on criteria on geographic location, vegetation type, climate and permafrost. This is a confusing description because one site groups can contain subregions from different parts of Asia, so it is hard to understand how they could have been defined on the basis of geographic location or climate, for example.

*Our response:* **We agree that the description in the manuscript was not very clear and we**

**have modified it in the new version. Site-groups were defined by pollen data with the same**

**vegetation-climate-permafrost conditions and similar pollen components and temporal**

**patterns. In the new version, we re-organized Figure 1 following the results of cluster**

**analysis to make the manuscript easier to read.**

Lines 165-170

*"Here, due to the sparse distribution of available sites, we divided the 203 sites into 42 site-groups, based on criteria of geographic location, vegetation type (vegetation zone map modified from Tseplyayev, 1961; Dulamsuren et al., 2005; Hou, 2001), climate (based on modern precipitation and temperature contours), and permafrost (Brown, 1997) following the strategy of Li (2016), and the pollen data within one site-group should be of similar components and temporal patterns."*

---

## Author Comment (AC3) · 31 Mar 2019

[revised manuscript text omitted]

---

## Author Response (AR2)

We hope that the reversion of this manuscript meets the requirements of Climate of the Past. If you have more comments, we will be very thankful and we are pleasure to improve our manuscript.

On behalf of all co-author.

All best wishes

Xianyong Cao

[revised manuscript text omitted]

---

## Author Response (AR3)

**Response to editor:**

**Dear Editor,**

   **Thank you very much for your comments to improve our manuscript. We have responded to the comments one by one and revised our manuscript accordingly. Sorry for the misunderstanding about the insignificant vegetation changes, in the new version, we have deleted the description about the insignificant results.**

**The major changes as follows:**

- include the citations of the data(sets) used in the main reference list (i.e. the publications for Appendix 2 need to be included in the manuscript and those for Appendix 4 need to be moved). Posting the references online is not sufficient as the original data producers are not credited for the appearance of a citation in a pangaea file (such citation don't count). You rely on their work (also for future syntheses), so they should be properly cited. Please see the instructions for manuscript preparation (https://www.climate-of-the-past.net/for_authors/manuscript_preparation.html), which explicitly mentions the proper citation of data sets.

**Response: Agree. We have presented these publications in the main reference list in the new version. Please see the tracked version of this manuscript.**

- I could not find a letter detailing the response to the reviewer and noticed that comment 6 (copied below) was not completely addressed, specifically wrt to the discussion. Please address this comment and provide a description/motivation of the changes together with your revised version. (6- Figure 3 and the related discussion. The groups that failed the broken-stick test are not assumed to be significant. Is that statistically correct to still use these insignificant results? Furthermore, do the authors have been using these groups to calculate the summary of frequency in figure 3? All of this should be specified and discuss (if data are used for discussion). I expect that if the broken-stick test failed, it does not mean that there is no changes but rather we do not have any thoughts about what is happening.)

**Response: We had used these site groups with insignificant vegetation changes to calculate the summary of frequency in figure 3. In the new version, we have deleted the description and discussion about the insignificant results and modified the Figure 3.**

Line 237-244:

"*Overall, the middle Holocene (including 8.5, 7.5, 6.5, and 5.5 ka time-slices) has the highest frequency of primary vegetation changes. Records from inland areas such as the West Siberian Plain, central Yakutia, and northern Mongolia are characterized by relatively many middle-Holocene splits. There are seven site-groups whose primary vegetation changes during the early Holocene (including 11.5, 10.5, and 9.5 ka time-slices), and most of them from the south-eastern coastal part of the study area. Only three site-groups have late-Holocene primary vegetation changes (Fig. 3).*"

Line 398-401:

"*PFT datasets from only 19 site-groups pass the broken-stick test for clustering analysis, and most of them have only one significant vegetation change, further supporting the case that only slight changes occurred during the Holocene in northern Asia.*"

And a few final grammar and typing changes. The line numbers below refer to the ms with tracked changes

-L101: change "into" to "to"

-L208: please reword the part in brackets

-L301: change "once" to "ones"

-L464: change "taxon" to "taxa"

-L527: unclear, please reword

-L581: change "an" to "a"

**Response: Agree. We have corrected these sentences in the new version. Please see the version with tracked changes.**